# Near-Optimal Sample Complexity for MDPs via Anchoring

**Jongmin Lee** [1]   **Mario Bravo** [2]   **Roberto Cominetti** [3]

## Abstract

We study a new model-free algorithm to compute $\varepsilon$-optimal policies for average reward Markov decision processes, in the weakly communicating setting. Given a generative model, our procedure combines a recursive sampling technique with Halpern's anchored iteration, and computes an $\varepsilon$-optimal policy with sample and time complexity $\widetilde{O}(|\mathcal{S}||\mathcal{A}|\|h^*\|_{\mathrm{sp}}^2/\varepsilon^2)$ both in high probability and in expectation. To our knowledge, this is the best complexity among model-free algorithms, matching the known lower bound up to a factor $\|h^*\|_{\mathrm{sp}}$. Although the complexity bound involves the span seminorm $\|h^*\|_{\mathrm{sp}}$ of the unknown bias vector, the algorithm requires no prior knowledge and implements a stopping rule which guarantees with probability 1 that the procedure terminates in finite time. We also analyze how these techniques can be adapted for discounted MDPs.

## 1. Introduction

A main task in reinforcement learning is to compute an $\varepsilon$-optimal policy for Markov Decision Processes (MDPs) when the transition probabilities are unknown. If one has access to a generative model that provides independent samples of the next state for any given initial state and action, a standard approach in *model-based* algorithms is to approximate the transition probabilities by sampling to a sufficiently high accuracy, and build a surrogate model which is then solved by dynamic programming techniques. In contrast, *model-free* methods recursively approximate a solution of the Bellman equation for the value function, with lower

---

[1]Department of Mathematical Sciences, Seoul National University, Seoul, Korea [2]Facultad de Ingeniería y Ciencias, Universidad Adolfo Ibáñez, Santiago, Chile [3]Institute for Mathematical & Computational Engineering and Department of Industrial and Systems Engineering, Pontificia Universidad Católica de Chile, Santiago, Chile. Correspondence to: Jongmin Lee <dlwhd2000@snu.ac.kr>, Mario Bravo <mario.bravo@uai.cl>, Roberto Cominetti <roberto.cominetti@uc.cl>.

*Proceedings of the $42^{nd}$ International Conference on Machine Learning*, Vancouver, Canada. PMLR 267, 2025. Copyright 2025 by the author(s).

memory requirements, specially when combined with function approximation techniques for large scale problems.

The sample complexity of model-free algorithms has been studied both in the discounted and the average reward setups. While for discounted MDPs these procedures have been shown to achieve optimal sample complexity, previous results for average rewards are less satisfactory and exhibit a substantial gap with respect to the known lower bound. Furthermore, most prior model-free and model-based algorithms for average reward MDPs require for their implementation some *a priori* bound on the mixing times or on the span seminorm of the bias vector.

### 1.1. Our contribution

We propose a new model-free algorithm to compute an $\varepsilon$-optimal policy for weakly communicating average reward MDPs. We show that, with probability at least $1 - \delta$, the algorithm achieves the task with a sample and time complexity of order $O(\widetilde{L}|\mathcal{S}||\mathcal{A}|\|h^*\|_{\mathrm{sp}}^2/\varepsilon^2)$. Here $\mathcal{S}$ and $\mathcal{A}$ are the state and action spaces of the MDP and $\|h^*\|_{\mathrm{sp}}$ is the span seminorm of the bias vector described later in Section 1.2, while $\widetilde{L} = \ln(|\mathcal{S}||\mathcal{A}|\|h^*\|_{\mathrm{sp}}/(\varepsilon\delta)) \ln^4(\|h^*\|_{\mathrm{sp}}/\varepsilon)$ is a logarithmic multiplicative factor. This yields a near-optimal algorithm since this complexity matches the known lower bound up to a factor $\|h^*\|_{\mathrm{sp}}$. More explicit estimates of the algorithm's complexity are discussed in Section 3.2.

Our algorithm can be implemented without any prior knowledge about $\|h^*\|_{\mathrm{sp}}$ nor other characteristics of the optimal policy. Also, it implements a stopping rule which guarantees with probability 1 that the method finds an $\varepsilon$-optimal policy in finite time, with the above complexity bound holding both in high probability as well as in expectation. This compares favorably to previous model-free and model-based methods which usually require some prior estimates of the span seminorm $\|h^*\|_{\mathrm{sp}}$, or an upper bound for the maximal mixing time $t_{\mathrm{mix}}$ (see Table 1 and Section 1.3). To our knowledge, the only algorithm with explicit complexity guarantees and which does not require prior knowledge is the model-based method in the recent paper by Tuynman et al. (2024), which however applies for communicating MDPs, achieving a complexity bound of $\widetilde{O}(|\mathcal{S}||\mathcal{A}| D/\varepsilon^2 + |\mathcal{S}|^2|\mathcal{A}|D^2)$, where $D$ stands for the diameter of the MDP.

The core of our analysis builds on the following key ideas.

| Previous works | Sample complexity | Method type | MDP class | Prior knowledge |
|---|---|---|---|---|
| Wang (2017) | $\widetilde{O}\left(|\mathcal{S}||\mathcal{A}|\tau^2 t_{\mathrm{mix}}^2/\varepsilon^2\right)$ | model-free | M | ◯ |
| Jin & Sidford (2020) | $\widetilde{O}\left(|\mathcal{S}||\mathcal{A}|t_{\mathrm{mix}}^2/\varepsilon^2\right)$ | model-free | M | ◯ |
| Jin & Sidford (2021) | $\widetilde{O}\left(|\mathcal{S}||\mathcal{A}|t_{\mathrm{mix}}/\varepsilon^3\right)$ | model-based | M | ◯ |
| Li et al. (2024) | $\widetilde{O}\left(|\mathcal{S}||\mathcal{A}|t_{\mathrm{mix}}^3/\varepsilon^2\right)$ | model-free | M | ◯ |
| Wang et al. (2024) | $\widetilde{O}\left(|\mathcal{S}||\mathcal{A}|t_{\mathrm{mix}}/\varepsilon^2\right)$ | model-based | M | ◯ |
| Wang et al. (2022) | $\widetilde{O}\left(|\mathcal{S}||\mathcal{A}|\|h^*\|_{\mathrm{sp}}/\varepsilon^3\right)$ | model-based | W | ◯ |
| Zhang & Xie (2023) | $\widetilde{O}\left(|\mathcal{S}||\mathcal{A}|\|h^*\|_{\mathrm{sp}}^2/\varepsilon^2\right)$ | model-free | W | ◯ |
| Zurek & Chen (2024) | $\widetilde{O}\left(|\mathcal{S}||\mathcal{A}|\|h^*\|_{\mathrm{sp}}/\varepsilon^2\right)$ | model-based | W | ◯ |
| Zurek & Chen (2024) | $\widetilde{O}\left(|\mathcal{S}||\mathcal{A}|(b_{\mathrm{tran}} + \|h^*\|_{\mathrm{sp}})/\varepsilon^2\right)$ | model-based | G | ◯ |
| Jin et al. (2024a) | $\widetilde{O}\left(|\mathcal{S}||\mathcal{A}|t_{\mathrm{mix}}^8/\varepsilon^8\right)$ | model-free | M | △ |
| Bravo & Contreras (2024) | $\widetilde{O}\left(|\mathcal{S}||\mathcal{A}|\|h^*\|_{\mathrm{sp}}^7/\varepsilon^7\right)$ | model-free | W | △ |
| Zurek & Chen (2025) | $\widetilde{O}\left(|\mathcal{S}||\mathcal{A}|\|h^*\|_{\mathrm{sp}}^2/\varepsilon^2\right)$ | model-based | W | △ |
| This work (Theorem 3.2) | $\widetilde{O}\left(|\mathcal{S}||\mathcal{A}|\|h^*\|_{\mathrm{sp}}^2/\varepsilon^2\right)$ | model-free | W | △ |
| Tuynman et al. (2024) | $\widetilde{O}\left(|\mathcal{S}||\mathcal{A}|D/\varepsilon^2\right)$ | model-based | C | ✕ |
| This work (Corollary 3.5) | $\widetilde{O}\left(|\mathcal{S}||\mathcal{A}|\|h^*\|_{\mathrm{sp}}^2/\varepsilon^2\right)$ | model-free | W | ✕ |

*Table 1.* **Algorithms for average reward MDPs.** The table shows the leading term in the sample complexity for computing an $\varepsilon$-optimal policy with probability at least $1-\delta$. The notation $\widetilde{O}(\cdot)$ hides logarithmic factors including $\log(1/\delta)$. The column 'MDP class' distinguishes the type of MDP to which the algorithm applies: (M) uniformly mixing or ergodic MDPs with finite mixing times $t_{\mathrm{mix}}$; (C) communicating MDPs with finite diameter $D$; (W) weakly communicating MDPs with bias vector $h^*$; and (G) general MDPs with transient parameter $b_{\mathrm{tran}}$. The symbol '◯' denotes an algorithm that requires prior knowledge of $t_{\mathrm{mix}}$, $\|h^*\|_{\mathrm{sp}}$, or $b_{\mathrm{tran}}$ for its implementation. Similarly, '△' stands for algorithms that run without prior knowledge but require it to fix the number of iterations or the number of samples. Finally '✕' applies to algorithms that do not require any prior knowledge. For a more thorough discussion see Section 1.3.

First, we show that any approximate solution of the Bellman equation with an $\varepsilon$ residual error yields an $\varepsilon$-optimal policy. To achieve an accurate empirical estimation of the Bellman residual error, we use a recursive sampling technique analyzed using an Azuma-Hoeffding-like inequality. Leveraging these estimates, we draw on concepts from the domain of fixed-points for nonexpansive maps, where the classical Halpern (or Anchored) iteration is known to be efficient (see Section 1.3 for a detailed discussion). This combined approach represents our main contribution and, to the best of our knowledge, is being applied for the first time to average reward MDPs in the generative setup. In the last section we adapt these ideas to study a similar algorithm for discounted MDPs and compare with other existing methods.

**Notations.** For a finite set $J$ we denote $\Delta(J)$ the set of probability distributions over $J$. Also, for $x \in \mathbb{R}^J$ we denote by $\|x\|_\infty = \max_{j\in J}|x(j)|$ its infinity norm and by $\|x\|_{\mathrm{sp}} = \max_{j\in J} x(j) - \min_{j\in J} x(j)$ its span seminorm. For $x \in \mathbb{R}^J$ and a real number $c \in \mathbb{R}$, we denote $x - c$ the vector with components $x(j) - c$ for all $j \in J$.

### 1.2. The model

Let us recall some basic facts about *average reward Markov decision processes*. These can be found in the classical ref-

erences (Bertsekas, 2012; Puterman, 2014; Sutton & Barto, 2018). We consider a Markov decision process $(\mathcal{S}, \mathcal{A}, \mathcal{P}, r)$ with finite state space $\mathcal{S}$, finite action space $\mathcal{A}$, transition kernel $\mathcal{P}\colon \mathcal{S} \times \mathcal{A} \to \Delta(\mathcal{S})$, and reward $r\colon \mathcal{S} \times \mathcal{A} \to \mathbb{R}$. Given an initial state $s_0 = s$ and a stationary and deterministic control policy $\pi\colon \mathcal{S} \to \mathcal{A}$, the average reward or gain $g_\pi \in \mathbb{R}^{\mathcal{S}}$ is defined by

$$g_\pi(s) = \lim_{n\to\infty} \mathbb{E}_\pi\left[\tfrac{1}{n}\textstyle\sum_{t=0}^{n-1} r(s_t, a_t) \mid s_0 = s\right]$$

where $\mathbb{E}_\pi$ denotes the expected value over all trajectories $(s_0, a_0, s_1, a_1, \ldots, s_t, a_t, \ldots)$ of the Markov chain induced by the control $a_t = \pi(s_t)$ with transitions $s_{t+1} \sim \mathcal{P}(s_t, a_t)$. The optimal reward is then $g^*(s) = \max_\pi g_\pi(s)$. A policy $\pi$ is called $\varepsilon$-optimal if $0 \le g^*(s) - g_\pi(s) \le \varepsilon$ for all $s \in \mathcal{S}$.

Although one could consider more elaborated history-dependent randomized policies $\pi$, when $\mathcal{S}$ and $\mathcal{A}$ are finite the optimum is already attained with the simpler deterministic stationary policies (Puterman, 2014), so we restrict to such policies $\pi$ and we denote $P_\pi$ the transition matrix of the induced Markov chain, namely, $P_\pi(s'|s) = \mathcal{P}(s'|s, \pi(s))$.

**Classification of MDPs.** An MDP is called *unichain* if the Markov chain $P_\pi$ induced by every policy $\pi$ has a single

recurrent class plus a possibly empty set of transient states[1]. The MDP is said to be *weakly communicating* if there is a set of states where each state in the set is accessible from every other state in that set under some policy, plus a possibly empty set of states that are transient for all policies. Otherwise, the MDP is called *multichain* in which case each $P_\pi$ may have multiple recurrent classes. Unichain MDPs are weakly communicating, while multichain is the weakest notion. In this paper we set our study in the middle ground by adopting the following standing assumption:

(H) *The MDP $(\mathcal{S}, \mathcal{A}, \mathcal{P}, r)$ is weakly communicating.*

**Bellman's equation.** An optimal policy $\pi$ can be obtained by first solving the following Bellman equation for the *gain* $g \in \mathbb{R}^{\mathcal{S}}$ and *bias* $h \in \mathbb{R}^{\mathcal{S}}$, namely: for all $s \in \mathcal{S}$

$$h(s) + g(s) = \max_{a \in \mathcal{A}} \Big\{ r(s,a) + \sum_{s' \in \mathcal{S}} \mathcal{P}(s'|s,a)\, h(s') \Big\},$$

and then taking actions $\pi(s) = a \in \mathcal{A}$ that attain the max. This Bellman equation always has solutions (Puterman, 2014, Corollary 9.1.5). Under the assumption (H) the gain $g(s) \equiv g^*$ is constant across all states with $g^*$ the optimal average reward (Puterman, 2014, Theorems 8.3.2, 8.4.1, 9.1.2, 9.1.6) so that hereafter we treat the gain $g^*(s)$ as a scalar $g^* \in \mathbb{R}$. Moreover, (Puterman, 2014, Theorems 9.1.7, 9.1.8) show that choosing actions that maximize the right hand side in the Bellman equation yields an optimal policy.

Bellman's equation can be rewritten in terms of the so-called $Q$-factors. Namely, denoting $Q(s,a)$ the expression inside the maximum, the system becomes: for all $(s,a) \in \mathcal{S} \times \mathcal{A}$

$$Q(s,a) + g^* = r(s,a) + \sum_{s' \in \mathcal{S}} \mathcal{P}(s'|s,a) \max_{a' \in \mathcal{A}} Q(s',a'),$$

and then an optimal policy can be obtained by selecting $\pi(s) \in \mathrm{argmax}_{a \in \mathcal{A}} Q(s,a)$.

This $Q$-version of the Bellman equation can be expressed more compactly as the fixed-point equation[2]

$$Q = r + \mathcal{P} \max_{\mathcal{A}}(Q) - g^*$$

where $\max_{\mathcal{A}} : \mathbb{R}^{\mathcal{S} \times \mathcal{A}} \to \mathbb{R}^{\mathcal{S}}$ assigns to each $Q \in \mathbb{R}^{\mathcal{S} \times \mathcal{A}}$ the vector $h \in \mathbb{R}^{\mathcal{S}}$ with $h(s) = \max_{a \in \mathcal{A}} Q(s,a)$, whereas $\mathcal{P} : \mathbb{R}^{\mathcal{S}} \to \mathbb{R}^{\mathcal{S} \times \mathcal{A}}$ transforms each $h \in \mathbb{R}^{\mathcal{S}}$ into the matrix $\mathcal{P}h \in \mathbb{R}^{\mathcal{S} \times \mathcal{A}}$ with $\mathcal{P}h(s,a) = \sum_{s' \in \mathcal{S}} \mathcal{P}(s'|s,a)h(s')$.

In what follows we denote $\mathcal{T} : \mathbb{R}^{\mathcal{S} \times \mathcal{A}} \to \mathbb{R}^{\mathcal{S} \times \mathcal{A}}$ the map $\mathcal{T}(Q) = r + \mathcal{P} \max_{\mathcal{A}}(Q)$ so that Bellman's equation reduces to $Q = \mathcal{T}_{g^*}(Q)$ with $\mathcal{T}_{g^*}(Q) = \mathcal{T}(Q) - g^*$. We denote $Q^*$ an arbitrary fixed-point, and $h^* = \max_{\mathcal{A}}(Q^*)$ the corresponding solution for the original form of Bellman's equation.

---

[1]For basic concepts in Markov chains such as recurrent classes, transient states, accessibility, irreducibility, etc., we refer to Puterman (2014, Appendix A.2).

[2]Recall that for a matrix $B \in \mathbb{R}^{\mathcal{S} \times \mathcal{A}}$ and a real number $g \in \mathbb{R}$, the operation $B - g$ subtracts $g$ from each component $B(s,a)$.

## 1.3. Previous work

**Average reward MDPs.** The setup of average reward MDPs was introduced in the dynamic programming literature by Howard (1960), while Blackwell (1962) established a theoretical framework for their analysis. Reinforcement learning provides a variety of methods to approximately solve average reward MDPs in the case where the transition matrix and reward are unknown (Mahadevan, 1996; Dewanto et al., 2020). These methods include model-based algorithms (Jin & Sidford, 2021; Zurek & Chen, 2024), model-free algorithms (Wei et al., 2020; Wan et al., 2021), policy gradient methods (Bai et al., 2024; Kumar et al., 2025), and mirror descent (Murthy & Srikant, 2023).

**Model-based methods with a generative model.** Model-based algorithms use sampling to estimate the transition probabilities of the MDP, and then solve the surrogate model by using standard dynamic programming techniques. These methods can achieve optimal sample complexity in both the discounted and average reward setups. For discounted MDPs, Agarwal et al. (2020); Li et al. (2020) achieve optimal sample complexity by matching the lower bound in Azar et al. (2013). For average reward MDPs satisfying a uniform mixing condition, Jin & Sidford (2021) proposed a model-based algorithm based on a reduction to a discounted MDP. Also using reductions to discounted MDPs, (Wang et al., 2022) relaxed the uniform mixing assumption to deal with weakly communicating MDPs; Wang et al. (2024) achieved optimal sample complexity assuming finite mixing times; and, Zurek & Chen (2024) achieved optimal complexity for weakly communicating and multichain MDPs.

**Model-free methods with a generative model.** Most model-free algorithms directly estimate the $Q$-factors and policy without learning a model of the transition probabilities. They are more efficient in terms of computation and memory requirements, and can tackle large scale problems when combined with function approximation.

In the generative model setup (Kearns & Singh, 1998) the sample complexity of model-free algorithms has been widely studied. For discounted MDPs they achieve optimal sample complexity matching the complexity lower bound (Sidford et al., 2018; Wainwright, 2019; Jin et al., 2024b). For average rewards, Table 1 presents a summary of the sample complexity of previous model-free and model-based algorithms. Specifically, for MDPs with finite mixing times $t_{\mathrm{mix}}$, Wang (2017) developed a model-free method that applies a primal-dual algorithm to a bilinear saddle point reformulation of the Bellman equation. Also under the mixing condition, Jin & Sidford (2020) use a stochastic mirror descent framework to solve bilinear saddle point problems, whereas Li et al. (2024) studied an actor-critic method also based on stochastic mirror descent. Zhang

& Xie (2023) first obtained sample complexity results for weakly communicating MDPs, by applying Q-learning with variance reduction to an approximation by a discounted MDP. Lower bounds on the sample complexity for mixing and weakly communicating MDPs, with orders of $t_{\mathrm{mix}}/\varepsilon^2$ and $\|h^*\|_{\mathrm{sp}}/\varepsilon^2$, respectively, were established by Jin & Sidford (2021) and Wang et al. (2022).

**Prior knowledge.** A drawback of these previous model-based and model-free algorithms is that their implementation requires prior estimates for the mixing times $t_{\mathrm{mix}}$ of the Markov chains induced by all policies, or the span seminorm $\|h^*\|_{\mathrm{sp}}$ of some solution of Bellman's equations. Several methods that partially overcome this limitation have been proposed recently. In the case of finite but unknown mixing times, Jin et al. (2024a) combine Q-learning and approximation by discounted MDPs with progressively larger discount factors, presenting a model-free algorithm with sample complexity of order $t_{\mathrm{mix}}^8/\varepsilon^8$. For weakly communicating MDPs, Bravo & Contreras (2024) develop a model-free value iteration that combines Halpern's anchoring with mini-batching, achieving a sample complexity of order $\|h^*\|_{\mathrm{sp}}^7/\varepsilon^7$. Also in the weakly communicating setting, Zurek & Chen (2025) recently proposed a model-based algorithm that uses a discounted MDP approximation with a sufficiently large discount factor, which is then solved by plugging in a generic subroutine, achieving a complexity of $\|h^*\|_{\mathrm{sp}}^2/\varepsilon^2$. Additionally, Neu & Okolo (2024) use a primal-dual stochastic gradient descent combined with a regularization technique, although the sample complexity is not stated in terms of the intrinsic characteristics of the MDP but in terms of the output of the algorithm, namely, using the expected value of the gain of the policy generated by the method.

Although the implementation of these methods does not involve explicitly the mixing time $t_{\mathrm{mix}}$, the diameter $D$, or the span seminorm $\|h^*\|_{\mathrm{sp}}$ of a bias vector, they do need an estimate of these quantities in order to fix the number of iterations that need to be run or the number of samples to collect in order to guarantee $\varepsilon$-optimality. In this sense, they fail to fully dispense for the need of prior knowledge. To the best of our knowledge, the only algorithms that can run without prior knowledge and guarantee at termination an $\varepsilon$-optimal policy are the model-based algorithm by Tuynman et al. (2024), which applies to communicating MDPs by estimating the diameter $D$, and the model-free algorithm SAVIA+ presented in this paper which applies to the larger class of weakly communicating MDPs by implementing an effective stopping rule based on the empirical Bellman residual. These latter algorithms achieve sample complexities of order $D/\varepsilon^2$ and $\|h^*\|_{\mathrm{sp}}^2/\varepsilon^2$ respectively.

**Value Iterations.** Value iterations (VIs)—an instantiation of the Banach-Picard fixed point iterations—were among the first methods considered in the dynamic programming literature (Bellman, 1957) and serve as a fundamental algorithm to compute the value function for discounted MDPs as well as unichain average reward MDPs. The sample-based variants, such as TD-Learning (Sutton, 1988), Fitted Value Iteration (Ernst et al., 2005; Munos & Szepesvári, 2008), and Deep Q-Network (Mnih et al., 2015), are the workhorses of modern reinforcement learning algorithms (Bertsekas & Tsitsiklis, 1996; Sutton & Barto, 2018; Szepesvári, 2010). VIs are also routinely applied in diverse settings, including factored MDPs (Rosenberg & Mansour, 2021), robust MDPs (Kumar et al., 2024), MDPs with reward machines (Bourel et al., 2023), and MDPs with options (Fruit et al., 2017). In the generative model setup, variance reduction sampling was applied to approximate VIs for discounted rewards: Sidford et al. (2023; 2018) use precomputed offsets to reduce variance of sampling, Wainwright (2019) applies SVRG-type variance reduction sampling (Johnson & Zhang, 2013) to Q-learning, and Jin et al. (2024b) use SARAH-type variance reduction sampling (Nguyen et al., 2017) which we also exploit in this work.

**Halpern iterations.** For $\gamma$-contractions on Banach spaces, the classical Banach-Picard iterates $x^{k+1} = T(x^k)$ converge to the unique fixed point $x^* = T(x^*)$, with explicit bounds for the residuals $\|T(x^k) - x^k\| \le \gamma^k \|T(x^0) - x^0\|$ and the distance $\|x^k - x^*\| \le \frac{\gamma^k}{1-\gamma}\|T(x^0) - x^0\|$ to the fixed point. This fits well for discounted MDPs, and is also useful in the average reward setting for unichain MDPs.

For nonexpansive maps with $\gamma = 1$, as it is the case for average reward MDPs, these estimates degenerate and provide no useful information. An alternative is provided by Halpern's iteration $x^{k+1} = (1 - \beta_{k+1})x^0 + \beta_{k+1}T(x^k)$, where the next iterate is computed as a convex combination between $T(x^k)$ and the initial point $x^0$ which acts as an *anchor* point along the iterations (Halpern, 1967). The sequence $\beta_k \in (0, 1)$ is chosen exogenously and increasing to 1, so that the strength of the anchor mechanism diminishes as the iteration progresses.

Halpern's anchored iteration has been widely studied in minimax optimization and fixed-point problems (Halpern, 1967; Sabach & Shtern, 2017; Lieder, 2021; Park & Ryu, 2022; Contreras & Cominetti, 2022; Yoon & Ryu, 2021; Cai et al., 2022). In the context of reinforcement learning, Lee & Ryu (2023; 2025) applied the anchoring technique to VIs achieving an accelerated convergence rate for cumulative-reward MDPs and the first non-asymptotic rate for average reward multichain MDPs. As mentioned, Bravo & Contreras (2024) applied the anchoring mechanism to Q-learning for average reward MDPs with a generative model.

Assuming that the set of fixed points $\mathrm{Fix}(T)$ is nonempty, and under suitable conditions on $\beta_k$, Halpern's iterates have

been proved to converge towards a fixed point in the case of Hilbert spaces (Wittmann, 1992) as well as in uniformly smooth Banach spaces (Reich, 1980; Xu, 2002). For more general normed spaces, the analysis in Sabach & Shtern (2017, Lemma 5) implies that for $\beta_k = k/(k + 2)$ one has the explicit error bound $\|T(x^k) - x^k\| \leq \frac{4}{k+1}\|x^0 - x^*\|$. The proportionality constant in this bound was recently improved in the Hilbert setting by Lieder (2021); Kim (2021). For a comprehensive analysis, including the determination of the optimal Halpern iteration, see Contreras & Cominetti (2022).

## 2. Framework overview

### 2.1. Anchored Value Iteration

As discussed in Section 1.2, Bellman's equation is equivalent to the fixed point equation $Q = \mathcal{T}_{g^\star}(Q)$ where $\mathcal{T}_{g^\star}(Q) = \mathcal{T}(Q) - g^\star$ with $\mathcal{T}(Q) = r + \mathcal{P}\max_{\mathcal{A}}(Q)$. Since $\max_{\mathcal{A}}$ and $\mathcal{P}$ are nonexpansive for the norm $\|\cdot\|_\infty$ in the corresponding spaces, the same holds for $\mathcal{T}_{g^*}$ and one might consider Halpern's iteration

$$Q^{k+1} = (1 - \beta_{k+1})Q^0 + \beta_{k+1}\mathcal{T}_{g^*}(Q^k). \qquad (1)$$

This recursion involves the unknown optimal value $g^*$. However, since $\mathcal{T}(Q + c) = \mathcal{T}(Q) + c$ is homogeneous by addition of constants $c \in \mathbb{R}$, one can interpret (1) in the quotient quotient space $\mathcal{M} = \mathbb{R}^{\mathcal{S} \times \mathcal{A}}/E$ with $E$ the subspace of constant matrices, and consider instead the *implementable* iteration where $g^*$ is removed

$$Q^{k+1} = (1 - \beta_{k+1})Q^0 + \beta_{k+1}\mathcal{T}(Q^k). \qquad \text{(Anc-VI)}$$

As explained in Appendix C, both (1) and (Anc-VI) are equivalent up to constants and can be interpreted as a standard Halpern iteration in the quotient space $\mathcal{M}$.

### 2.2. Reducing both Bellman residual and policy error

From the previous observations, it follows that the error bounds for Halpern's iteration directly translate into error bounds for (Anc-VI) in span seminorm. In particular, Sabach & Shtern (2017, Lemma 5) implies that for $\beta_k = k/(k + 2)$ (Anc-VI) the Bellman residual error converge to zero with the explicit bound in span seminorm

$$\|Q^k - \mathcal{T}(Q^k)\|_{\mathrm{sp}} \leq \frac{4}{k+1}\|Q^0 - Q^*\|_{\mathrm{sp}}$$

where $Q^*$ is any solution of Bellman's equation. Moreover, for weakly communicating MDPs we can prove that any $Q$ with a small residual $\|Q - \mathcal{T}(Q)\|_{\mathrm{sp}}$ yields an approximately optimal policy. More precisely, adapting (Puterman, 2014, Theorem 9.1.7), we have following result.

**Proposition 2.1.** *Let $Q \in \mathbb{R}^{\mathcal{S} \times \mathcal{A}}$ and $\pi : \mathcal{S} \to \mathcal{A}$ a greedy policy such that $\pi(s) \in \mathrm{argmax}_{a \in \mathcal{A}} Q(s, a)$. Then for all states $s \in \mathcal{S}$ we have $0 \leq g^* - g_\pi(s) \leq \|Q - \mathcal{T}(Q)\|_{\mathrm{sp}}$.*

Combining Proposition 2.1 with the general estimate of the Bellman residual error, it follows that

$$g^* - g_{\pi_k}(s) \leq \|Q^k - \mathcal{T}(Q^k)\|_{\mathrm{sp}} \leq \frac{4}{k+1}\|Q^0 - Q^*\|_{\mathrm{sp}}$$

where $\pi_k(s) \in \mathrm{argmax}_{a \in \mathcal{A}} Q^k(s, a)$. Thus (Anc-VI) not only reduces the Bellman residual error but it allows to derive $\varepsilon$-optimal policies.

### 2.3. Estimating $\mathcal{T}(Q^k)$ by recursive sampling

We are interested in the generative setting where the transition kernel $\mathcal{P}$ is not known but one can generate samples from $\mathcal{P}(\cdot|s, a)$. In this case (Anc-VI) cannot be implemented since one cannot compute $\mathcal{T}(Q^k)$. With a generative model, a natural approach is to approximate $T^k \approx \mathcal{T}(Q^k)$ by computing $h^k = \max_{\mathcal{A}}(Q^k)$ and then collecting samples $\{s_j\}_{j=1}^{m_k} \sim \mathcal{P}(\cdot|s, a)$ in order to set

$$T^k(s, a) = r(s, a) + \frac{1}{m_k}\sum_{j=1}^{m_k} h^k(s_j).$$

We note that the number of samples required to attain a small error scales with the norm of $h^k$. In order to reduce the sample complexity, we use instead a *recursive sampling* technique borrowed from Jin et al. (2024b). The basic idea is to exploit the previous approximation $T^{k-1} \approx \mathcal{T}(Q^{k-1})$ and the linearity of the map $\mathcal{P}$, and to approximate $\mathcal{T}(Q^k)$ by estimating the difference $\mathcal{T}(Q^k) - \mathcal{T}(Q^{k-1}) = \mathcal{P}d^k$ with $d^k = h^k - h^{k-1}$ and adding it to $T^{k-1}$. Specifically, for each $(s, a)$ take $m_k$ samples $s_j \sim \mathcal{P}(\cdot|s, a)$ and set

$$T^k(s, a) = T^{k-1}(s, a) + \frac{1}{m_k}\sum_{j=1}^{m_k} d^k(s_j), \quad k \geq 0.$$

Starting with $T^{-1} = r \in \mathbb{R}^{\mathcal{S} \times \mathcal{A}}$ and $h^{-1} = 0 \in \mathbb{R}^{\mathcal{S}}$, and denoting $D^k \approx \mathcal{P}d^k$ the matrix sampled at the $k$-th stage, we have

$$T^k = r + \sum_{i=0}^{k} D^i \qquad (2)$$

so that all the previous estimates $\{D_i\}_{i=0}^{k}$ are used to approximate $\mathcal{T}(Q^k)$. The subsequent analysis will show that $\|d^k\|_{\mathrm{sp}}$ can be significantly smaller than $\|h^k\|_{\mathrm{sp}}$, achieving a significant reduction in the overall sample complexity.

Our algorithms, to be presented in next section, will use as a subroutine the following generic sampling procedure.

---
**Algorithm 1** SAMPLE$(d, m)$

---
**Input:** $d \in \mathbb{R}^{\mathcal{S}}$ ; $m \in \mathbb{N}$
**for** $(s, a) \in \mathcal{S} \times \mathcal{A}$ **do**
$\quad D(s, a) = \frac{1}{m}\sum_{j=1}^{m} d(s_j)$ with $s_j \overset{iid}{\sim} \mathcal{P}(\cdot \mid s, a)$
**end for**
**Output:** $D$

---

For later reference we observe that this subroutine draws $|\mathcal{S}||\mathcal{A}|\, m$ samples and its output satisfies $\|D\|_{\mathrm{sp}} \leq \|d\|_{\mathrm{sp}}$. In

what follows we focus on the sample complexity, that is to say, the total number of samples required by the algorithms. If we assume that each sample requires constant $O(1)$-time, then the time complexity of the algorithms will be of the same order as the sampling complexity.

## 3. Stochastic Anchored Value Iteration

We may now present our method which combines two basic ingredients: an anchored value iteration and recursive sampling. The following SAVIA iteration is our basic algorithm which considers a fixed sampling sequence $c_k > 0$ and an averaging sequence $\beta_k \in (0, 1)$ increasing to 1.

---

**Algorithm 2** SAVIA$(Q^0, n, \varepsilon, \delta)$

---

**Input:** $Q^0 \in \mathbb{R}^{\mathcal{S} \times \mathcal{A}}$ ; $n \in \mathbb{N}$ ; $\varepsilon > 0$ ; $\delta \in (0, 1)$
$\alpha = \ln(2|\mathcal{S}||\mathcal{A}|(n+1)/\delta)$
$T^{-1} = r$ ; $h^{-1} = 0$ ; $\beta_0 = 0$
**for** $k = 0, \ldots, n$ **do**
    $Q^k = (1 - \beta_k)\, Q^0 + \beta_k\, T^{k-1}$
    $h^k = \max_{\mathcal{A}}(Q^k)$
    $d^k = h^k - h^{k-1}$
    $m_k = \max\{\lceil \alpha\, c_k \|d^k\|_{\mathrm{sp}}^2 / \varepsilon^2 \rceil, 1\}$
    $D^k = \mathrm{SAMPLE}(d^k, m_k)$
    $T^k = T^{k-1} + D^k$
**end for**
$\pi^n(s) \in \mathrm{argmax}_{a \in \mathcal{A}} Q^n(s, a) \quad (\forall s \in \mathcal{S})$
**Output:** $(Q^n, T^n, \pi^n)$

---

The iteration can start from an arbitrary $Q^0$, including the all-zero matrix. However, if one has a prior estimate $Q^0 \approx Q^*$ (*e.g.* from a similar MDP solved previously), it may be convenient to start from $Q^0$ since our complexity bounds scale with the distance $\|Q^0 - Q^*\|_{\mathrm{sp}}$. In any case we stress that SAVIA does not require any prior knowledge and all the parameters in the algorithm are independent of $Q^*$ or $h^*$, including the sequences $\beta_k, c_k$ for which we provide a specific choice in the next subsection.

### 3.1. Sample complexity for the basic SAVIA iteration.

In order to study the sample complexity of SAVIA we first establish conditions which ensure that with high probability the recursive sampling provides good approximations $T^k \approx \mathcal{T}(Q^k)$ for all $k$'s. We only present the main results and ideas, and defer all proofs to the Appendix.

**Proposition 3.1.** *Let $c_k > 0$ with $2\sum_{k=0}^{\infty} c_k^{-1} \leq 1$ and $T^k, Q^k$ the iterates generated by SAVIA$(Q^0, n, \varepsilon, \delta)$. Then, with probability at least $1 - \delta$ we have $\|T^k - \mathcal{T}(Q^k)\|_{\infty} \leq \varepsilon$ simultaneously for all $k = 0, \ldots, n$.*

The proof is an adaptation of the arguments leading to the Azuma-Hoeffding inequality (Azuma, 1967), by consider-

ing the specific choice of number of samples $m_k$ in the recursive sampling, combined with some ad-hoc union bounds.

REMARK. A direct consequence of Proposition 3.1 is that, with probability at least $1 - \delta$, the true Bellman residual error $\|Q^k - \mathcal{T}(Q^k)\|_{\mathrm{sp}}$ and the empirical residual $\|Q^k - T^k\|_{\mathrm{sp}}$ differ at most by $2\varepsilon$, as results from a triangular inequality for $\|\cdot\|_{\mathrm{sp}}$ and the general estimate $\|\cdot\|_{\mathrm{sp}} \leq 2\|\cdot\|_{\infty}$.

In what follows we consider the specific sequences

$$\text{(S)} \quad \begin{cases} c_k = 5(k+2)\ln^2(k+2) \\ \beta_k = k/(k+2) \end{cases}$$

which satisfy $2\sum_{k=0}^{\infty} c_k^{-1} \leq 1$ and $\beta_k$ increasing to 1. These $\beta_k$'s are the same as in Sabach & Shtern (2017), and provide explicit guarantees for the reduction of the Bellman residual error. The choice for $c_k$ has been carefully tailored to achieve a small sample complexity.

The following result presents our complexity bound for SAVIA for weakly communicating MDPs satisfying (H).

**Theorem 3.2.** *Assume* (H) *and* (S) *and let* $(Q^n, T^n, \pi^n)$ *be the output computed by* SAVIA$(Q^0, n, \varepsilon, \delta)$. *Then, with probability at least* $(1 - \delta)$ *we have, for all* $s \in \mathcal{S}$

$$g^* - g_{\pi^n}(s) \leq \|Q^n - \mathcal{T}(Q^n)\|_{\mathrm{sp}} \leq \frac{8\|Q^0 - Q^*\|_{\mathrm{sp}}}{n+2} + 4\varepsilon,$$

*with a sample and time complexity of order*

$$O\big(L\,|\mathcal{S}||\mathcal{A}|\big((\|Q^0 - Q^*\|_{\mathrm{sp}}^2 + \|Q^0\|_{\mathrm{sp}}^2)/\varepsilon^2 + n^2\big)\big)$$

*where* $L = \ln(|\mathcal{S}||\mathcal{A}|(n+1)/\delta)\ln^3(n+2)$.

The proof exploits Proposition 3.1 and the remark above in order to establish an upper bound of the Bellman residual error. The bound for the optimality gap in the policy error then follows from Proposition 2.1. The recursive sampling technique is crucial here to attain a complexity that scales quadratically in $\varepsilon$, whereas a naive sampling would give a much worse order complexity. Note that $L$ contains only logarithmic factors and is bounded away from 0 so that in our complexity estimates it can absorb any constant factors.

### 3.2. A model-free algorithm without prior knowledge

Theorem 3.2 shows that SAVIA is effective in reducing the Bellman residual error as well as the policy error. Namely, with probability at least $1 - \delta$, after $n \geq \|Q^0 - Q^*\|_{\mathrm{sp}}/\varepsilon$ iterations it produces a $(12\,\varepsilon)$-optimal policy $\pi^n$ and with $\widetilde{O}\big(|\mathcal{S}||\mathcal{A}|(\|Q^0 - Q^*\|_{\mathrm{sp}}^2/\varepsilon^2 + 1)\big)$ complexity.

Unfortunately, since $Q^*$ is unknown we cannot directly determine the number of iterations $n$ required to achieve this goal. To bypass this issue, we modify the basic iteration by running SAVIA for an increasing sequence of $n$'s using a standard doubling trick (Auer et al., 1995; Besson

& Kaufmann, 2018), and incorporating an explicit stopping rule based on the empirical Bellman residual error $\|Q^{n_i} - T^{n_i}\|_{\mathrm{sp}}$. Specifically, we consider the SAVIA+ iteration described in Algorithm 3.

---

**Algorithm 3** SAVIA+$(Q^0, \varepsilon, \delta)$

---
**Input:** $Q^0 \in \mathbb{R}^{\mathcal{S} \times \mathcal{A}}$ ; $\varepsilon > 0$ ; $\delta \in (0, 1)$
**for** $i = 0, 1, \ldots$ **do**
  Set $n_i = 2^i, \delta_i = \delta/c_i$.
  $(Q^{n_i}, T^{n_i}, \pi^{n_i}) = \text{SAVIA}(Q^0, n_i, \varepsilon, \delta_i)$
**until** $\|Q^{n_i} - T^{n_i}\|_{\mathrm{sp}} \le 14\varepsilon$
**Output:** $Q^{n_i}, T^{n_i}, \pi^{n_i}$

---

Our next results establish the effectiveness of the stopping rule and determine the sample complexity of the modified algorithm SAVIA+. In order to simplify the notation, and to avoid equation display issues, we express our estimates in terms of the following quantities

$$\mu = \|Q^0 - Q^*\|_{\mathrm{sp}},$$
$$\nu = \|Q^0 - Q^*\|_{\mathrm{sp}} + \|Q^0\|_{\mathrm{sp}},$$
$$\kappa = \max\{\|r\|_{\mathrm{sp}}, \|Q^0\|_{\mathrm{sp}}\}.$$

Define the stopping time of SAVIA+ as the random variable

$$N = \inf\{n_i \in \mathbb{N} : \|Q^{n_i} - T^{n_i}\|_{\mathrm{sp}} \le 14\varepsilon\}$$

where $Q^{n_i}, T^{n_i}$'s are the iterates generated in each loop. We first show that $N$ has finite expectation and, as a consequence, that the algorithm stops in finite time.

**Proposition 3.3.** *Assume* (H) *and* (S). *Then,* $\mathbb{E}[N] \le 2(1 + \mu/\varepsilon)/(1 - \delta)$. *In particular $N$ is finite almost surely and* SAVIA+$(Q^0, \varepsilon, \delta)$ *stops with probability* 1 *after finitely many loops.*

The proof exploits the fact that restarting SAVIA from $Q^0$ in every cycle guarantees the independence of the stopping events in each loop. This independence is also relevant to establish the following sample complexity of SAVIA+, which is one of the main results of this paper.

**Theorem 3.4.** *Assume* (H) *and* (S). *Let* $(Q^N, T^N, \pi^N)$ *be the output of* SAVIA+$(Q^0, \varepsilon, \delta)$. *Then, with probability at least* $(1 - \delta)$ *we have, for all* $s \in \mathcal{S}$

$$g^* - g_{\pi^N}(s) \le \|Q^N - \mathcal{T}(Q^N)\|_{\mathrm{sp}} \le 16\varepsilon,$$

*with sample and time complexity* $O\big(\widehat{L}\,|\mathcal{S}||\mathcal{A}|(\nu^2/\varepsilon^2 + 1)\big)$ *where* $\widehat{L} = \ln\big(|\mathcal{S}||\mathcal{A}|\,(1 + \mu/\varepsilon)/\delta\big)\ln^4(1 + \mu/\varepsilon)$.

In the proof, we consider some 'good' events $G_i$ (see Appendix) which guarantee a low sample complexity for the $i$-th loop and then, through simple union bounds, we show that the probability of these good events occurring at every iteration is at least $1 - \delta$. Thus Theorem 3.4 shows that

SAVIA+ computes an $\varepsilon$-optimal policy without requiring any prior estimates for the number of iteration to run, thanks to the doubling trick and the stopping rule.

In order to compare our complexity result with the lower bound and the previous results in Table 1, which concern the case $\|h^*\|_{\mathrm{sp}} \ge 1$, we state the following Corollary under this assumption. Note however that Theorem 3.4 holds without any additional restriction.

**Corollary 3.5.** *Assume* (H), (S), $r(s, a) \in [0, 1]$ *for all* $(s, a) \in \mathcal{S} \times \mathcal{A}$, *and* $\|h^*\|_{\mathrm{sp}} \ge 1$. *Let* $(Q^N, T^N, \pi^N)$ *be the output of* SAVIA+$(Q^0, \varepsilon/16, \delta)$ *with* $Q^0 = 0$ *and* $\varepsilon \le 1$. *Then, with probability at least* $(1 - \delta)$ *we have, for all* $s \in \mathcal{S}$

$$g^* - g_{\pi^N}(s) \le \|Q^N - \mathcal{T}(Q^N)\|_{\mathrm{sp}} \le \varepsilon,$$

*with sample and time complexity* $O\big(\widetilde{L}\,|\mathcal{S}||\mathcal{A}|\|h^*\|_{\mathrm{sp}}^2/\varepsilon^2\big)$ *where* $\widetilde{L} = \ln\big(|\mathcal{S}||\mathcal{A}|\|h^*\|_{\mathrm{sp}}/(\varepsilon\delta)\big)\ln^4(\|h^*\|_{\mathrm{sp}}/\varepsilon)$.

The proof is a straightforward application of Theorem 3.4 considering that $\|Q^*\|_{\mathrm{sp}} \le \|r\|_{\mathrm{sp}} + \|h^*\|_{\mathrm{sp}}$. To the best of our knowledge, the sample complexity in Corollary 3.5 is state-of-the-art among model-free algorithms, and matches the complexity lower bound up to a factor $\|h^*\|_{\mathrm{sp}}$. Furthermore, SAVIA+ does not require any prior knowledge of $\|h^*\|_{\mathrm{sp}}$ and automatically provides an $\varepsilon$-optimal policy, unlike most prior works.

Our previous results provide PAC bounds (probably approximately correct) which guarantee a small error with high probability. On the other hand, it is also relevant to estimate the expected sample complexity considering the full probability space, including the low probability events in which the algorithm may take a significantly longer time to converge. Such expected complexity results have been studied mainly in the multi-armed bandit literature (Katz-Samuels & Jamieson, 2020; Mason et al., 2020; Jourdan et al., 2023), but much less in the reinforcement learning literature. The following result establishes the expected sample complexity of SAVIA+ in terms of expected policy error.

**Theorem 3.6.** *Assume* (H) *and* (S). *Let* $(Q^N, T^N, \pi^N)$ *be the output of* SAVIA+$(Q^0, \varepsilon, \delta)$. *Then, for all* $s \in \mathcal{S}$

$$\mathbb{E}[g^* - g_{\pi^N}(s)] \le 16\varepsilon + \delta\|r\|_{\mathrm{sp}},$$

*with expected sample and time complexity*

$$\widetilde{O}\big(|\mathcal{S}||\mathcal{A}|(\nu^2/\varepsilon^2 + 1 + \delta\,(1 + \mu/\varepsilon)^2(1 + (\kappa/\varepsilon)^2))\big).$$

In contrast with the proof of Theorem 3.4 which only focuses on the occurrence of good events, the analysis in expectation requires to estimate the sample complexity for the 'unlucky' events where Theorem 3.2 does not provide a guarantee of low sample complexity. To this end we establish a uniform bound for $\|d^k\|_{\mathrm{sp}} \le \max\{\|r\|_{\mathrm{sp}}^2, \|Q^0\|_{\mathrm{sp}}^2\}$ and compute the expected sample complexity considering

both the lucky and unlucky events by the law of total expectation.

Observe that, compared to Theorem 3.4, the expected policy error and expected sample complexity include the additional terms $\delta \|r\|_{\mathrm{sp}}$ and $\widetilde{O}(|\mathcal{S}||\mathcal{A}|\delta\,(1+\mu/\varepsilon)^2(1+(\kappa/\varepsilon)^2))$ respectively. These additional terms arise from the the unlucky events, showing a fourth order in $\varepsilon$ while our PAC complexity is quadratic in $\varepsilon$.

From Theorem 3.6 we derive following analog of Corollary 3.5 for the expected policy error and complexity.

**Corollary 3.7.** *Assume* (H), (S), $r(s,a) \in [0,1]$ *for all* $(s,a) \in \mathcal{S} \times \mathcal{A}$ *and* $\|h^*\|_{\mathrm{sp}} \geq 1$. *Let* $(Q^N, T^N, \pi^N)$ *be the output of* SAVIA+$(Q^0, \varepsilon/17, \delta)$ *with* $Q^0 = 0$, $\varepsilon \leq 1$, *and* $\delta = \varepsilon^2/17$. *Then, for all* $s \in \mathcal{S}$

$$\mathbb{E}[g^* - g_{\pi^N}(s)] \leq \varepsilon,$$

*with expected sample complexity* $\widetilde{O}\left(|\mathcal{S}||\mathcal{A}|\|h^*\|_{\mathrm{sp}}^2/\varepsilon^2\right)$.

Interestingly, by setting $\delta = \varepsilon^2/17$, the expected sample complexity of Corollary 3.7 has same order as Corollary 3.5. We are not aware of such estimates on the expected complexity for MDPs with a generative model.

# 4. Discounted MDPs

In this section, we extend SAVIA to the discounted setup. We will show that with minor changes the approach applies to the discounted case, using essentially the same key ideas.

## 4.1. The model

Let us recall the setup. We consider a discounted Markov decision process $(\mathcal{S}, \mathcal{A}, \mathcal{P}, r, \gamma)$, where $\gamma \in (0,1)$ is the discount factor. The assumptions on the state and action spaces, transition probabilities, and rewards remain the same as in the average reward case, except for condition (H) which is no longer required. Given an initial state $s_0 = s$ and action $a_0 = a$ and a stationary and deterministic policy $\pi \colon \mathcal{S} \to \mathcal{A}$, the $Q$-value function is now defined by

$$Q_\pi(s,a) = \mathbb{E}_\pi[\textstyle\sum_{t=0}^\infty \gamma^t r(s_t, a_t) \,|\, s_0 = s, a_0 = a]$$

where $\mathbb{E}_\pi$ denotes the expected value over all trajectories $(s_0, a_0, s_1, a_1, \ldots, s_t, a_t, \ldots)$ induced by $\mathcal{P}$ and $\pi$.

The optimal $Q$-value is $Q^*(s,a) = \max_\pi Q_\pi(s,a)$ and an optimal policy chooses $\pi(s) \in \mathrm{argmax}_{a \in \mathcal{A}} Q^*(s,a)$. A policy $\pi$ is called $\varepsilon$-optimal if $\|Q^* - Q_\pi\|_\infty \leq \varepsilon$.

It is well-known that $Q^*$ is the unique solution of the following the Bellman equation, for all $(s,a) \in \mathcal{S} \times \mathcal{A}$

$$Q(s,a) = r(s,a) + \gamma \sum_{s' \in \mathcal{S}} \mathcal{P}(s'|s,a) \max_{a' \in \mathcal{A}} Q(s',a').$$

Similarly to the case of average rewards, by introducing the $\gamma$-contracting map $\mathcal{T}_\gamma : \mathbb{R}^{\mathcal{S} \times \mathcal{A}} \to \mathbb{R}^{\mathcal{S} \times \mathcal{A}}$ defined as $\mathcal{T}_\gamma(Q) = r + \gamma \mathcal{P} \max_{\mathcal{A}}(Q)$, this is equivalent to the fixed point equation $Q = \mathcal{T}_\gamma(Q)$.

## 4.2. Sample complexity of SAVID

We present now our algorithms SAVID and SAVID+ adapted to the setting of discounted MDPs. The main difference is the introduction of the discount factor in the sampling process updates

$$T^k = T^{k-1} + \gamma D^k = r + \gamma \textstyle\sum_{i=0}^k D^i$$

and the fact that we measure errors using the infinity norm instead of the span seminorm. Also, the previous variable $h$ is now called $V$, to reflect the nature of the value function in this setting. With these premises, all the elements of our approach, including the anchored value iteration, recursive sampling, and proof techniques, are directly adapted to the discounted setup.

---

**Algorithm 4** SAVID$(Q^0, n, \varepsilon, \delta, \gamma)$

---

**Input:** $Q^0 \in \mathbb{R}^{\mathcal{S} \times \mathcal{A}}$; $n \in \mathbb{N}$; $\varepsilon > 0$; $\delta \in (0,1)$
$\quad \alpha = \ln(2|\mathcal{S}||\mathcal{A}|(n+1)/\delta)$
$\quad T^{-1} = r$; $V^{-1} = 0$; $\beta_0 = 0$
$\quad$**for** $k = 0, \ldots, n$ **do**
$\quad\quad Q^k = (1 - \beta_k)\, Q^0 + \beta_k\, T^{k-1}$
$\quad\quad V^k = \max_{\mathcal{A}}(Q^k)$
$\quad\quad d^k = V^k - V^{k-1}$
$\quad\quad m_k = \max\{\lceil 2\,\alpha\,c_k\,\|d^k\|_\infty^2 / \varepsilon^2\rceil, 1\}$
$\quad\quad D^k = \mathrm{SAMPLE}(d^k, m_k)$
$\quad\quad T^k = T^{k-1} + \gamma D^k$
$\quad$**end for**
$\quad \pi^n(s) \in \mathrm{argmax}_{a \in \mathcal{A}} Q^n(s,a) \quad (\forall s \in \mathcal{S})$
**Output:** $Q^n, T^n, \pi^n$

---

---

**Algorithm 5** SAVID+$(Q^0, \varepsilon, \delta, \gamma)$

---

**Input:** $Q^0 \in \mathbb{R}^{\mathcal{S} \times \mathcal{A}}$; $\varepsilon > 0$; $\delta \in (0,1)$
$\quad$**for** $i = 0, 1, \ldots$ **do**
$\quad\quad$Set $n_i = 2^i, \delta_i = \delta/c_i$.
$\quad\quad (Q^{n_i}, T^{n_i}, \pi^{n_i}) = $ SAVID$(Q^0, n_i, \varepsilon, \delta_i, \gamma)$
$\quad$**until** $\|Q^{n_i} - T^{n_i}\|_\infty \leq 11\,\varepsilon$
**Output:** $Q^{n_i}, T^{n_i}, \pi^{n_i}$

---

The following result connects the Bellman residual error to the policy error. Compared to Proposition 2.1, there is $1 - \gamma$ loss when we translate the Bellman residual error $\|Q - \mathcal{T}_\gamma(Q)\|_\infty$ into a policy error.

**Proposition 4.1.** *Let* $Q \in \mathbb{R}^{\mathcal{S} \times \mathcal{A}}$ *and* $\pi : \mathcal{S} \to \mathcal{A}$ *a greedy policy such that* $\pi(s) \in \mathrm{argmax}_{a \in \mathcal{A}} Q(s,a)$. *Then we have* $\|Q^* - Q_\pi\|_\infty \leq \frac{2}{1-\gamma} \|Q - \mathcal{T}_\gamma(Q)\|_\infty$.

Recall that a major difficulty in the average reward setting was the fact that in general we do not have an *a priori* bound on the optimal bias vector $h^*$. For discounted MDPs, we have the simple bound $\|Q^*\|_\infty \leq \|r\|_\infty / (1-\gamma)$, that can be used to run SAVID and which allows us to obtain the following sample complexity.

**Theorem 4.2.** *Assume* (S)*, $r(s,a) \in [0,1]$ for all $(s,a) \in \mathcal{S} \times \mathcal{A}$, and $n = \lceil 10/((1-\gamma)\varepsilon) \rceil$. Let $(Q^n, T^n, \pi^n)$ be the output of* SAVID$(Q^0, n, \epsilon/10, \delta, \gamma)$ *with $Q^0 = 0$ and $\varepsilon \leq 1/(1-\gamma)$. Then, with probability at least $(1-\delta)$ we have*

$$\|Q^n - \mathcal{T}_\gamma(Q^n)\|_\infty \leq \varepsilon,$$

*with sample and time complexity $O\left(L_\gamma |\mathcal{S}||\mathcal{A}|/((1-\gamma)^2\varepsilon^2)\right)$ where $L_\gamma = \ln(2|\mathcal{S}||\mathcal{A}|/((1-\gamma)\varepsilon\delta))\ln^3(2/((1-\gamma)\varepsilon))$.*

It is known that for discounted MDPs the lower bound on the complexity to compute an $\varepsilon$-optimal $Q$-value function and an $\varepsilon$-optimal policy is $\widetilde{\Omega}(|\mathcal{S}||\mathcal{A}|/((1-\gamma)^3\varepsilon^2))$ (Azar et al., 2013). In fact, model-free algorithms can achieve this complexity; see (Wainwright, 2019) for $Q$-values and (Sidford et al., 2018) for optimal policies. Since $\|Q - \mathcal{T}_\gamma(Q)\|_\infty \leq (1+\gamma)\|Q^* - Q\|_\infty$, it follows that these algorithms require $\widetilde{O}(|\mathcal{S}||\mathcal{A}|/((1-\gamma)^3\varepsilon^2))$ to obtain an $\varepsilon$ Bellman residual error. Up to our knowledge, the $\widetilde{O}(|\mathcal{S}||\mathcal{A}|)/((1-\gamma)^2\varepsilon^2))$ sample complexity in Theorem 4.2 is the best known sample complexity to obtain an $\varepsilon$ residual error for discounted MDPs.

On the other hand, Proposition 4.1 implies that, to compute an $\varepsilon$-optimal policy with arbitrary high probability, SAVID requires $\widetilde{O}(|\mathcal{S}||\mathcal{A}|/((1-\gamma)^4\varepsilon^2))$ sample calls, matching the lower bound up to a factor $1/(1-\gamma)$. In what follows, we show that SAVID**+** can improve this upper bound, making the dependence explicit on $\|Q^*\|_\infty^2$ and saving a factor of $1/(1-\gamma)^2$ by using a doubling trick and a stopping rule.

### 4.3. Sample complexity of SAVID+

**Theorem 4.3.** *Assume* (S)*, $r(s,a) \in [0,1]$ for all $(s,a) \in \mathcal{S} \times \mathcal{A}$, and $\|Q^*\|_\infty \geq 1$. Let $(Q^N, T^N, \pi^N)$ be the output of* SAVID+$(Q^0, \varepsilon(1-\gamma)/24, \delta, \gamma)$ *with $Q^0 = 0$ and $\varepsilon \leq 1/(1-\gamma)$. Then, with probability at least $(1-\delta)$ we have*

$$\|Q^* - Q_{\pi^N}\|_\infty \leq 2\|Q^N - \mathcal{T}_\gamma(Q^N)\|_\infty /(1-\gamma) \leq \varepsilon,$$

*with sample and time complexity*

$$O\left(\widetilde{L}_\gamma |\mathcal{S}||\mathcal{A}| \|Q^*\|_\infty^2 /((1-\gamma)^2\varepsilon^2)\right)$$

*where $\widetilde{L}_\gamma = \ln\left(2|\mathcal{S}||\mathcal{A}|/((1-\gamma)\varepsilon\delta)\right)\ln^4\left(2/((1-\gamma)\varepsilon)\right)$.*

Notice that if we use the bound $\|Q^*\|_\infty \leq 1/(1-\gamma)$ we get $\widetilde{O}(|\mathcal{S}||\mathcal{A}|/((1-\gamma)^4\varepsilon^2))$ complexity, which matches the one of SAVID. However, the sample complexity of SAVID**+** shows an explicit dependence on $\|Q^*\|_\infty$ that might be useful for particular MDPs. For instance, if we

have $\|Q^*\|_\infty = O(1/\sqrt{1-\gamma})$, the complexity bound in Theorem 4.3 improves to $\widetilde{O}(|\mathcal{S}||\mathcal{A}|/((1-\gamma)^3\varepsilon^2))$. Note however that the worst case example in (Azar et al., 2013) is such that $\|Q^*\|_\infty = \Omega(1/(1-\gamma))$.

## 5. Conclusion

This work proposed a novel framework Stochastic Anchored Value Iteration that computes an $\varepsilon$-optimal policy with anchored value iteration and recursive sampling. Our model-free algorithm SAVIA**+** does not require prior knowledge of the bias vector and achieves near-optimal sample complexity for weakly communicating average reward MDPs, matching the lower bound up to a factor $\|h^*\|_{sp}$. Similarly, SAVID**+** attains near-optimal complexity for discounted MDPs.

A possible research direction is to improve the sample complexity of SAVIA**+** to match the lower bound for weakly communicating MDPs. Other interesting open questions are the analysis of the anchoring framework for general multi-chain MDPs, and the study of alternative sampling setups such as episodic sampling and online learning.

## Acknowledgments

The work of Mario Bravo and Roberto Cominetti was partially supported by FONDECYT 1241805. Jongmin Lee was supported by the Samsung Science and Technology Foundation (Project Number SSTF-BA2101-02).

## Impact Statement

This paper focuses on the theoretical aspects of reinforcement learning. There are no societal impacts that we anticipate from our theoretical result.

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

# A. Omitted proofs for the average reward setting

In this section, we present the proofs omitted in the main body of the paper, along with additional comments to complement our results. We start by analyzing the average reward setup, while the analog results in the discounted setting is presented in Appendix B. For better readability, we restate the main results.

## A.1. Proof of Proposition 2.1

**Proposition 2.1.** *Let $Q \in \mathbb{R}^{\mathcal{S} \times \mathcal{A}}$ and $\pi : \mathcal{S} \to \mathcal{A}$ a policy such that $\pi(s) \in \operatorname{argmax}_{a \in \mathcal{A}} Q(s, a)$. Then for all states $s \in \mathcal{S}$ we have $0 \leq g^* - g_\pi(s) \leq \|Q - \mathcal{T}(Q)\|_{\mathrm{sp}}$.*

*Proof.* Let $P_\pi$ the transition matrix of the Markov chain induced by $\pi$, that is, $P_\pi(s'|s) = \mathcal{P}(s'|s, \pi(s))$. Then $g_\pi = P_\pi^\infty r_\pi$ where $P_\pi^\infty = \lim_{T \to \infty} \frac{1}{T} \sum_{t=0}^{T-1} (P_\pi)^t$ and $r_\pi(s) = r(s, \pi(s))$. Since $P_\pi^\infty = P_\pi^\infty P_\pi$, it follows that for all $h \in \mathbb{R}^{\mathcal{S}}$ we have $g_\pi = P_\pi^\infty (r_\pi + P_\pi h - h)$. In particular if we take $h = \max_{\mathcal{A}}(Q)$ so that $h(s) = Q(s, \pi(s))$, then for every $s' \in \mathcal{S}$ the term $r_\pi(s') + (P_\pi h)(s') - h(s')$ is exactly the component $(s', \pi(s'))$ of the matrix $\mathcal{T}(Q) - Q$, and therefore by averaging according to $P_\pi^\infty(\cdot|s)$ we derive the inequality $g_\pi(s) \geq \min_{s', a'} \big(\mathcal{T}(Q) - Q\big)(s', a')$.

Similarly, for an optimal policy $\pi^*$ we have $g^* = P_{\pi^*}^\infty (r_{\pi^*} + P_{\pi^*} h - h)$ for $h = \max_{\mathcal{A}}(Q)$. Denoting $h'(s) = Q(s, \pi^*(s))$ we have $h'(s) \leq h(s)$ and therefore

$$r_{\pi^*}(s') + (P_{\pi^*} h)(s') - h(s') \leq r_{\pi^*}(s') + (P_{\pi^*} h)(s') - h'(s').$$

Noting that $r_{\pi^*}(s') + (P_{\pi^*} h)(s') - h'(s')$ is the component $(s', \pi^*(s'))$ of $\mathcal{T}(Q) - Q$, and averaging with $P_{\pi^*}^\infty(\cdot|s)$ we get $g^* = g^*(s) \leq \max_{s', a'} \big(\mathcal{T}(Q) - Q\big)(s', a')$. Sustracting both estimates we conclude $0 \leq g^* - g_\pi(s) \leq \|\mathcal{T}(Q) - Q\|_{\mathrm{sp}}$. $\square$

## A.2. Proofs of Section 3.1

Let $\mathcal{F}_k = \sigma(\{D^i : i = 0, \ldots, k\})$ denote the natural filtration generated by the sampling process in SAVIA, and $\mathbb{P}(\cdot)$ the probability distribution over the trajectories $(D^k)_{k \in \mathbb{N}}$. Notice that, because of the order of the updates, $T^k$ is $\mathcal{F}_k$-measurable whereas $Q^k$, $h^k$, $d^k$ and $m_k$, being functions of $T^{k-1}$, are $\mathcal{F}_{k-1}$-measurable.

**Proposition 3.1.** *Let $c_k > 0$ with $2 \sum_{k=0}^\infty c_k^{-1} \leq 1$ and $T^k, Q^k$ the iterates generated by $\mathrm{SAVIA}(Q^0, n, \varepsilon, \delta)$. Then, with probability at least $1 - \delta$ we have $\|T^k - \mathcal{T}(Q^k)\|_\infty \leq \varepsilon$ simultaneously for all $k = 0, \ldots, n$.*

*Proof.* Let $Y^i = D^i - \mathcal{P} d^i$ and $X^k = \sum_{i=0}^k Y^i$. Since $h^{-1} = 0$, by telescoping $\mathcal{T}(Q^k) = r + \mathcal{P} h^k = r + \sum_{i=0}^k \mathcal{P} d^i$ and then using (2) we get $T^k - \mathcal{T}(Q^k) = \sum_{i=0}^k (D^i - \mathcal{P} d^i) = X^k$.

We proceed to estimate $\mathbb{P}(\|X^k\|_\infty \geq \varepsilon)$ by adapting the arguments of Azuma-Hoeffding's inequality. Let $s_{k,j}^{s,a} \sim \mathcal{P}(\cdot \mid s, a)$ for $j = 1, \ldots, m_k$ be the samples at the $k$-th iteration for $(s, a) \in \mathcal{S} \times \mathcal{A}$, so that

$$Y^k(s, a) = \frac{1}{m_k} \sum_{j=1}^{m_k} \big(d^k(s_{k,j}^{s,a}) - \mathcal{P} d^k(s, a)\big) \qquad \forall (s, a) \in \mathcal{S} \times \mathcal{A}.$$

Since $d^k$ and $m_k$ are $\mathcal{F}_{k-1}$-measurable, it follows that $\mathbb{E}[Y^k | \mathcal{F}_{k-1}] = 0$ and therefore $X^k$ is an $\mathcal{F}_k$-martingale. From Markov's inequality and the tower property of conditional expectations we get that for each $(s, a) \in \mathcal{S} \times \mathcal{A}$ and $\lambda > 0$

$$
\begin{aligned}
\mathbb{P}(X^k(s, a) \geq \varepsilon) &\leq e^{-\lambda \varepsilon} \, \mathbb{E}\big[\exp(\lambda \, X^k(s, a))\big] \\
&= e^{-\lambda \varepsilon} \, \mathbb{E}\big[\exp(\lambda \, X^{k-1}(s, a)) \, \mathbb{E}[\exp(\lambda \, Y^k(s, a)) \, | \, \mathcal{F}_{k-1}]\big].
\end{aligned}
\tag{3}
$$

Now, conditionally on $\mathcal{F}_{k-1}$, the terms $Y_j^k = \frac{1}{m_k} \big(d^k(s_{k,j}^{s,a}) - \mathcal{P} d^k(s, a)\big)$ in the sum of $Y^k(s, a)$ are independent random variables with zero mean and $|Y_j^k| \leq \frac{1}{m_k} \|d^k\|_{\mathrm{sp}}$ so that Hoeffding's Lemma gives us

$$\mathbb{E}\big[\exp(\lambda \, Y^k(s, a)) \, | \, \mathcal{F}_{k-1}\big] = \prod_{j=1}^{m_k} \mathbb{E}\big[\exp(\lambda \, Y_j^k) \, | \, \mathcal{F}_{k-1}\big] \leq \exp\big(\tfrac{1}{2} \lambda^2 \|d^k\|_{\mathrm{sp}}^2 / m_k\big). \tag{4}$$

Using (3) and (4), together with $m_k \geq \alpha \, c_k \|d^k\|_{\mathrm{sp}}^2 / \varepsilon^2$, a simple induction yields

$$\mathbb{E}[\exp(\lambda \, X^k(s, a))] \leq \exp\big(\tfrac{1}{2} \lambda^2 \varepsilon^2 \textstyle\sum_{i=0}^k c_i^{-1} / \alpha\big).$$

Then, since $\sum_{i=0}^{\infty} c_i^{-1} \leq \frac{1}{2}$ we get $\mathbb{P}(X^k(s,a) \geq \varepsilon) \leq \exp\left(-\lambda\varepsilon + \frac{1}{4}\lambda^2\varepsilon^2/\alpha\right)$ and taking $\lambda = 2\alpha/\varepsilon$ we deduce

$$\mathbb{P}(X^k(s,a) \geq \varepsilon) \leq \exp(-\alpha) = \delta/(2|\mathcal{S}||\mathcal{A}|(n+1)).$$

A symmetric argument yields the same bound for $\mathbb{P}(X^k(s,a) \leq -\varepsilon)$ so that $\mathbb{P}(|X^k(s,a)| \geq \varepsilon) \leq \delta/(|\mathcal{S}||\mathcal{A}|(n+1))$. Applying a union bound over all $(s,a) \in \mathcal{S} \times \mathcal{A}$ we get $\mathbb{P}(\|X^k\|_\infty \geq \varepsilon) \leq \delta/(n+1)$, and then a second union bound over $k$ gives $\mathbb{P}(\bigcup_{k=0}^{n}\{\|X^k\|_\infty \geq \varepsilon\}) \leq \delta$. The conclusion follows by taking the complementary event. $\square$

From now on we consider the specific sequences $c_k = 5(k+2)\ln^2(k+2)$ and $\beta_k = k/(k+2)$. We also fix some solution $Q^*$ of Bellman's equation $Q^* = \mathcal{T}(Q^*) - g^*$, and we set $h^* = \max_\mathcal{A}(Q^*)$. We keep the notation for all the sequences generated by SAVIA and we assume throughout that

$$\left\|T^k - \mathcal{T}(Q^k)\right\|_\infty \leq \varepsilon \qquad \text{for all } k = 0, \ldots, n, \tag{5}$$

which, in view of Proposition 3.1, holds with probability at least $(1-\delta)$. Also, as noted earlier, for $D = \text{SAMPLE}(d,m)$ we have $\|D\|_{\text{sp}} \leq \|d\|_{\text{sp}}$, which combined with the nonexpansivity of the map $Q \mapsto \max_\mathcal{A}(Q)$ implies

$$\|T^k - T^{k-1}\|_{\text{sp}} = \|D^k\|_{\text{sp}} \leq \|d^k\|_{\text{sp}} = \|h^k - h^{k-1}\|_{\text{sp}} \leq \|Q^k - Q^{k-1}\|_{\text{sp}}. \tag{6}$$

As a preamble to the proof of Theorem 3.2 we establish two preliminary technical Lemmas.

**Lemma A.1.** *Assuming* (S) *and* (5)*, we have that*

$$\|Q^k - Q^*\|_{\text{sp}} \leq \|Q^0 - Q^*\|_{\text{sp}} + \frac{2}{3}\varepsilon k, \quad \text{for all } k = 0, \ldots, n.$$

*Proof.* From the iteration $Q^k = (1 - \beta_k)Q^0 + \beta_k T^{k-1}$ with $\beta_k = \frac{k}{k+2}$ we get

$$\|Q^k - Q^*\|_{\text{sp}} \leq \frac{2}{k+2}\|Q^0 - Q^*\|_{\text{sp}} + \frac{k}{k+2}\|T^{k-1} - Q^*\|_{\text{sp}}. \tag{7}$$

Using the invariance of $\|\cdot\|_{\text{sp}}$ by addition of constants and the nonexpansivity of $\mathcal{T}$ for this seminorm, a triangle inequality together with $\|\cdot\|_{\text{sp}} \leq 2\|\cdot\|_\infty$ and the bound (5) imply

$$\|T^{k-1} - Q^*\|_{\text{sp}} = \|T^{k-1} - \mathcal{T}(Q^*)\|_{\text{sp}} \leq 2\varepsilon + \|Q^{k-1} - Q^*\|_{\text{sp}} \tag{8}$$

which plugged back into (7) yields

$$\|Q^k - Q^*\|_{\text{sp}} \leq \frac{2}{k+2}\|Q^0 - Q^*\|_{\text{sp}} + \frac{k}{k+2}\left(2\varepsilon + \|Q^{k-1} - Q^*\|_{\text{sp}}\right).$$

Denoting $\theta_k = (k+1)(k+2)\|Q^k - Q^*\|_{\text{sp}}$ this becomes $\theta_k \leq \theta_0(k+1) + 2\varepsilon k(k+1) + \theta_{k-1}$, and inductively

$$\begin{aligned}\theta_k &\leq \theta_0 \sum_{i=1}^{k}(i+1) + 2\varepsilon \sum_{i=1}^{k} i(i+1) + \theta_0 \\ &= \theta_0 \tfrac{1}{2}(k+1)(k+2) + \tfrac{2}{3}\varepsilon k(k+1)(k+2).\end{aligned}$$

The conclusion then follows dividing by $(k+1)(k+2)$. $\square$

**Lemma A.2.** *Assume* (S) *and* (5) *and let* $\rho_k = 2\|Q^0 - Q^*\|_{\text{sp}} + \frac{2}{3}\varepsilon k$. *Then*

$$\|Q^k - Q^{k-1}\|_{\text{sp}} \leq \frac{2}{k(k+1)} \sum_{i=1}^{k} \rho_{i+2} \quad \text{for all } k = 1, \ldots, n.$$

*Proof.* From $Q^k = \frac{2}{k+2}Q^0 + \frac{k}{k+2}T^{k-1}$ and $Q^{k-1} = \frac{2}{k+1}Q^0 + \frac{k-1}{k+1}T^{k-2}$ we derive

$$Q^k - Q^{k-1} = \frac{2}{(k+1)(k+2)}(T^{k-1} - Q^0) + \frac{k-1}{k+1}(T^{k-1} - T^{k-2}). \tag{9}$$

Now, (8) together with Lemma A.1 readily imply $\|T^{k-1} - Q^0\|_{\text{sp}} \leq \|T^{k-1} - Q^*\|_{\text{sp}} + \|Q^* - Q^0\|_{\text{sp}} \leq \rho_{k+2}$, and therefore using (6) we get

$$\|Q^k - Q^{k-1}\|_{\text{sp}} \leq \frac{2}{(k+1)(k+2)}\rho_{k+2} + \frac{k-1}{k+1}\|Q^{k-1} - Q^{k-2}\|_{\text{sp}}.$$

Denoting $\tilde{\theta}_k = k(k+1)\|Q^k - Q^{k-1}\|_{\text{sp}}$ we have $\tilde{\theta}_k \leq \frac{2k}{k+2}\rho_{k+2} + \tilde{\theta}_{k-1} \leq 2\rho_{k+2} + \tilde{\theta}_{k-1}$. Hence $\tilde{\theta}_k \leq 2\sum_{i=1}^{k}\rho_{i+2}$ from which the result follows directly. $\square$

**Theorem 3.2.** *Assume* (H) *and* (S) *and let* $(Q^n, T^n, \pi^n)$ *be the output computed by* SAVIA$(Q^0, n, \varepsilon, \delta)$. *Then, with probability at least* $(1 - \delta)$ *we have, for all* $s \in \mathcal{S}$

$$g^* - g_{\pi^n}(s) \le \|Q^n - \mathcal{T}(Q^n)\|_{\mathrm{sp}} \le \frac{8\|Q^0 - Q^*\|_{\mathrm{sp}}}{n + 2} + 4\varepsilon,$$

*with a sample and time complexity of order*

$$O\big(L\,|\mathcal{S}||\mathcal{A}|\big((\|Q^0 - Q^*\|_{\mathrm{sp}}^2 + \|Q^0\|_{\mathrm{sp}}^2)/\varepsilon^2 + n^2\big)\big)$$

*where* $L = \ln(|\mathcal{S}||\mathcal{A}|(n + 1)/\delta)\ln^3(n + 2)$.

*Proof.* From Proposition 3.1, with probability at least $(1 - \delta)$ we have $\|T^k - \mathcal{T}(Q^k)\|_{\mathrm{sp}} \le \varepsilon$ for all $k = 0, \dots, n$. Now, the recursion $Q^n = \frac{2}{n+2}Q^0 + \frac{n}{n+2}T^{n-1}$ implies

$$Q^n - \mathcal{T}(Q^n) = \frac{2}{n+2}\big(Q^0 - \mathcal{T}(Q^n)\big) + \frac{n}{n+2}\big(T^{n-1} - \mathcal{T}(Q^{n-1})\big) + \frac{n}{n+2}\big(\mathcal{T}(Q^{n-1}) - \mathcal{T}(Q^n)\big). \tag{10}$$

Using the fact that $Q^* = \mathcal{T}(Q^*) - g^*$ and the invariance of $\|\cdot\|_{\mathrm{sp}}$ by additive constants, a triangle inequality and the $\|\cdot\|_{\mathrm{sp}}$-nonexpansivity of $\mathcal{T}(\cdot)$ together with Lemma A.1, imply

$$\|Q^0 - \mathcal{T}(Q^n)\|_{\mathrm{sp}} \le \|Q^0 - Q^*\|_{\mathrm{sp}} + \|Q^* - Q^n\|_{\mathrm{sp}} \le \rho_n,$$

while Lemma A.2 gives $\|\mathcal{T}(Q^{n-1}) - \mathcal{T}(Q^n)\|_{\mathrm{sp}} \le \frac{2}{n(n+1)}\sum_{i=1}^n \rho_{i+2}$. Thus, applying a triangle inequality to (10) and using these estimates together with Proposition 2.1 and $\|\cdot\|_{\mathrm{sp}} \le 2\|\cdot\|_\infty$ we obtain

$$\begin{aligned}
g^* - g_\pi(s) \le \|Q^n - \mathcal{T}(Q^n)\|_{\mathrm{sp}} &\le \tfrac{2}{n+2}\rho_n + 2\varepsilon + \tfrac{2}{(n+1)(n+2)}\sum_{i=1}^n \rho_{i+2} \\
&= \tfrac{4(1+2n)}{(n+1)(n+2)}\|Q^0 - Q^*\|_{\mathrm{sp}} + \tfrac{4(3+8n+3n^2)}{3(n+1)(n+2)}\varepsilon \\
&\le \tfrac{8\|Q^0 - Q^*\|_{\mathrm{sp}}}{n+2} + 4\varepsilon.
\end{aligned}$$

To estimate the complexity, for $k \ge 1$, we use the inequality $\|d^k\|_{\mathrm{sp}} \le \|Q^k - Q^{k-1}\|_{\mathrm{sp}}$ in (6) and Lemma A.2 to find

$$\begin{aligned}
\|d^k\|_{\mathrm{sp}} &\le \tfrac{2}{k(k+1)}\sum_{i=1}^k \rho_{i+2} \\
&= \tfrac{4}{k+1}\|Q^0 - Q^*\|_{\mathrm{sp}} + \tfrac{2(k+5)}{3(k+1)}\varepsilon \\
&\le \tfrac{4}{k+1}\|Q^0 - Q^*\|_{\mathrm{sp}} + 2\varepsilon.
\end{aligned}$$

Now, to estimate the total number of samples $|\mathcal{S}||\mathcal{A}|\sum_{k=0}^n m_k$ we recall that $m_k = \max\{\lceil \alpha c_k \|d^k\|_{\mathrm{sp}}^2/\varepsilon^2\rceil, 1\}$ which can be bounded as $m_k \le 1 + \alpha c_k \|d^k\|_{\mathrm{sp}}^2/\varepsilon^2$. Then

$$\begin{aligned}
\sum_{k=0}^n m_k &\le (n + 1) + \tfrac{\alpha}{\varepsilon^2}\sum_{k=0}^n c_k \|d^k\|_{\mathrm{sp}}^2 \\
&\le (n + 1) + \tfrac{10\alpha}{\varepsilon^2}\ln^2(2)\|Q_0\|_{\mathrm{sp}}^2 + \tfrac{5\alpha}{\varepsilon^2}\sum_{k=1}^n (k + 2)\ln^2(k + 2)\big(\tfrac{4}{k+1}\|Q^0 - Q^*\|_{\mathrm{sp}} + 2\varepsilon\big)^2 \\
&\le (n + 1) + \tfrac{10\alpha}{\varepsilon^2}\ln^2(2)\|Q_0\|_{\mathrm{sp}}^2 + \sum_{k=1}^n \tfrac{240\alpha}{\varepsilon^2(k+1)}\ln^2(k + 2)\|Q^0 - Q^*\|_{\mathrm{sp}}^2 + 40\alpha\sum_{k=1}^n (k + 2)\ln^2(k + 2) \\
&= O\big(\alpha\|Q^0\|_{\mathrm{sp}}^2/\varepsilon^2 + \alpha\ln^3(n + 2)\|Q^0 - Q^*\|_{\mathrm{sp}}^2/\varepsilon^2 + \alpha n^2\ln^2(n + 2)\big), \tag{11}
\end{aligned}$$

where the third inequality results by using the trivial bounds $(a + b)^2 \le 2a^2 + 2b^2$ and $\frac{k+2}{k+1} \le \frac{3}{2}$, and the last equality from integral estimations of the sums. The announced complexity bound then follows by multiplying this estimate by $|\mathcal{A}||\mathcal{S}|$ and using the definition of $\alpha$ and $L$. $\qquad \square$

REMARK. The complexity analysis above can be refined to obtain an explicit multiplicative constant in $O(\cdot)$.

### A.3. Proofs of Section 3.2

We now proceed to establish the finite convergence and complexity of the algorithm SAVIA+, for which we introduce some additional notation. Let us recall the definition of the parameters

$$\mu = \|Q^0 - Q^*\|_{\text{sp}},$$
$$\nu = \|Q^0 - Q^*\|_{\text{sp}} + \|Q^0\|_{\text{sp}},$$
$$\kappa = \max\{\|r\|_{\text{sp}}, \|Q^0\|_{\text{sp}}\}.$$

The stopping time of SAVIA+ is the random variable

$$N = \inf\{n_i \in \mathbb{N} : \|Q^{n_i} - T^{n_i}\|_{\text{sp}} \le 14\,\varepsilon\}$$

with $T^{n_i}$ and $Q^{n_i}$'s the iterates generated in each loop of SAVIA+$(Q^0, \varepsilon, \delta)$. For notational convenience we also define

$$I = \inf\{i \in \mathbb{N} : \|Q^{n_i} - T^{n_i}\|_{\text{sp}} \le 14\,\varepsilon\}$$

so that in fact $N = 2^I$. We let $i_0 \in \mathbb{N}$ be the smallest integer satisfying $n_{i_0} \ge \|Q^0 - Q^*\|_{\text{sp}}/\varepsilon = \mu/\varepsilon$, so that either $i_0 = 0$ and $n_{i_0} = 1$ or $n_{i_0-1} = n_{i_0}/2 < \mu/\varepsilon$, which combined imply $n_{i_0} \le 2(1 + \mu/\varepsilon)$.

In order to estimate the sample complexity, we consider the random events

$$S_i = \{\|Q^{n_i} - T^{n_i}\|_{\text{sp}} \le 14\,\varepsilon\} \quad \text{and} \quad G_i = \{\|T^k - \mathcal{T}(Q^k)\|_\infty \le \varepsilon, \ \forall k = 0, \ldots, n_i\}$$

with $T^k$ and $Q^k$'s the inner iterates generated during the execution of SAVIA$(Q^0, n_i, \varepsilon, \delta_i)$ in the $i$-th loop of SAVIA+, and denote by $M_i$ the number of samples used during this call, so that the total sample complexity is $M \triangleq \sum_{i=0}^{I} M_i$. Observe that the $M_i$'s and $M$ are random variables.

With these preliminary definitions we proceed to establish the following simple but useful preliminary estimate.

**Lemma A.3.** *Assume* (H) *and* (S). *Then, for all* $i \ge i_0$ *we have* $\mathbb{P}(S_i) \ge \mathbb{P}(G_i) \ge 1 - \delta_i$.

*Proof.* Proposition 3.1 guarantees $\mathbb{P}(G_i) \ge 1 - \delta_i$ so it suffices to show that $G_i \subseteq S_i$. This follows from Theorem 3.2 since for $i \ge i_0$ and all $\omega \in G_i$ we have

$$\|Q^{n_i}(\omega) - T^{n_i}(\omega)\|_{\text{sp}} \le \|Q^{n_i}(\omega) - \mathcal{T}(Q^{n_i})(\omega)\|_{\text{sp}} + \|\mathcal{T}(Q^{n_i})(\omega) - T^{n_i}(\omega)\|_{\text{sp}}$$
$$\le \frac{8\|Q^0 - Q^*\|_{\text{sp}}}{n_i + 2} + 4\,\varepsilon + 2\,\varepsilon$$
$$\le 14\,\varepsilon. \qquad \square$$

**Proposition 3.3.** *Assume* (H) *and* (S). *Then,* $\mathbb{E}[N] \le 2(1 + \mu/\varepsilon)/(1 - \delta)$. *In particular* $N$ *is finite almost surely and* SAVIA+$(Q^0, \varepsilon, \delta)$ *stops with probability* 1 *after finitely many loops.*

*Proof.* Since in each loop SAVIA+$(Q^0, n_i, \varepsilon, \delta_i)$ restarts afresh from $Q^0$, the events $\{S_i : i \in \mathbb{N}\}$ are mutually independent and therefore

$$\mathbb{P}(I = i) = \mathbb{P}(\bigcap_{j=0}^{i-1} S_j^c \cap S_i) = \prod_{j=0}^{i-1} \mathbb{P}(S_j^c) \cdot \mathbb{P}(S_i).$$

From Lemma A.3 we get $\mathbb{P}(S_i^c) \le \mathbb{P}(G_i^c) \le \delta_i$ for all $i \ge i_0 + 1$ and then $\mathbb{P}(I = i) \le \prod_{j=i_0}^{i-1} \delta_j$. On the other hand, from $2\sum_{i=0}^{\infty} c_i^{-1} \le 1$ it follows that $\delta_j = \delta/c_j \le \delta/2$ and therefore $\mathbb{P}(I = i) \le (\delta/2)^{i-i_0}$. Using this estimate and the identity $n_i = n_{i_0} 2^{i-i_0}$, we obtain

$$\mathbb{E}[N] = \sum_{i=0}^{\infty} n_i \, \mathbb{P}(N = n_i)$$
$$\le n_{i_0} + \sum_{i=i_0+1}^{\infty} n_{i_0} 2^{i-i_0} \, \mathbb{P}(I = i)$$
$$\le n_{i_0}(1 + \sum_{i=i_0+1}^{\infty} \delta^{i-i_0}).$$

This last expression is exactly $n_{i_0}/(1 - \delta)$ and the conclusion follows using the bound $n_{i_0} \le 2(1 + \mu/\varepsilon)$. $\qquad \square$

**Theorem 3.4.** *Assume* (H) *and* (S). *Let* $(Q^N, T^N, \pi^N)$ *be the output of* SAVIA+$(Q^0, \varepsilon, \delta)$. *Then, with probability at least* $(1 - \delta)$ *we have, for all* $s \in \mathcal{S}$

$$g^* - g_{\pi^N}(s) \le \|Q^N - \mathcal{T}(Q^N)\|_{\mathrm{sp}} \le 16\,\varepsilon$$

*with sample and time complexity* $O\big(\widehat{L}\,|\mathcal{S}||\mathcal{A}|(\nu^2/\varepsilon^2 + 1)\big)$ *where* $\widehat{L} = \ln\big(|\mathcal{S}||\mathcal{A}|\,(1 + \mu/\varepsilon)/\delta\big)\,\ln^4(1 + \mu/\varepsilon)$.

*Proof.* The inequality between the policy error and the Bellman error follows from Proposition 2.1 so it suffices to prove the second one. Consider the events $A = \{I \le i_0\}$ and $B = \bigcap_{i=0}^{\infty} G_i$. We claim that $\mathbb{P}(A \cap B) \ge 1 - \delta$. Indeed, by Proposition 3.1 we have $\mathbb{P}(G_i^c) \le \delta_i$ so that

$$\mathbb{P}(B^c) = \mathbb{P}(\textstyle\bigcup_{i=1}^{\infty} G_i^c) \le \textstyle\sum_{i=1}^{\infty} \delta/c_i \le \delta/2,$$

while Lemma A.3 implies $\mathbb{P}(A) \ge \mathbb{P}(S_{i_0}) \ge \mathbb{P}(G_{i_0}) \ge 1 - \delta_{i_0} \ge 1 - \delta/2$, which combined yield $\mathbb{P}(A^c \cup B^c) \le \delta$. Now, from the definition of $N$, $I$ and $G_i$, it follows that on the event $A \cap B$ we have

$$\|Q^N - \mathcal{T}(Q^N)\|_{\mathrm{sp}} \le \|Q^N - T^N\|_{\mathrm{sp}} + \|T^N - \mathcal{T}(Q^N)\|_{\mathrm{sp}} + \le 14\,\varepsilon + 2\,\varepsilon = 16\,\varepsilon.$$

Moreover, using (11) in the proof of Theorem 3.2, we can to bound the total sample complexity $M \triangleq \sum_{i=0}^{I} M_i$ as

$$M \le \textstyle\sum_{i=0}^{i_0} M_i = O\left(|\mathcal{S}||\mathcal{A}| \textstyle\sum_{i=0}^{i_0} \alpha_i \|Q^0\|_{\mathrm{sp}}^2/\varepsilon^2 + \alpha_i \ln^3(n_i + 2)\|Q^0 - Q^*\|_{\mathrm{sp}}^2/\varepsilon^2 + \alpha_i\,n_i^2 \ln^2(n_i + 2)\right),$$

where $\alpha_i = \ln(2|\mathcal{S}||\mathcal{A}|(n_i + 1)c_i/\delta)$ is the parameter used in the $i$-th internal cycle of SAVIA+. Note that $(n_i + 1)c_i \le 14n_i^2$ and therefore $\alpha_i \le \ln(28|\mathcal{S}||\mathcal{A}|n_i^2/\delta)$. Now, since the terms in the sum above increase with $i$, using the estimate $n_{i_0} \le 2(1 + \mu/\varepsilon)$ we conclude

$$\begin{aligned}
M &\le |\mathcal{S}||\mathcal{A}|\,\alpha_{i_0}(i_0 + 1)\,O\left(\|Q^0\|_{\mathrm{sp}}^2/\varepsilon^2 + \ln^3(n_{i_0} + 2)\|Q^0 - Q^*\|_{\mathrm{sp}}^2/\varepsilon^2 + n_{i_0}^2 \ln^2(n_{i_0} + 2)\right) \\
&\le |\mathcal{S}||\mathcal{A}|\,\alpha_{i_0} \ln^4(n_{i_0} + 2)\,O\left(\|Q^0\|_{\mathrm{sp}}^2/\varepsilon^2 + \mu^2/\varepsilon^2 + 4(1 + \mu/\varepsilon)^2\right) \\
&= O\big(\widehat{L}\,|\mathcal{S}||\mathcal{A}|\,O(\nu^2/\varepsilon^2 + 1)\big).
\end{aligned}$$
$\qquad\square$

**Corollary 3.5.** *Assume* (H), (S), $r(s, a) \in [0, 1]$ *for all* $(s, a) \in \mathcal{S} \times \mathcal{A}$, *and* $\|h^*\|_{\mathrm{sp}} \ge 1$. *Let* $(Q^N, T^N, \pi^N)$ *be the output of* SAVIA+$(Q^0, \varepsilon/16, \delta)$ *with* $Q^0 = 0$ *and* $\varepsilon \le 1$. *Then, with probability at least* $(1 - \delta)$ *we have, for all* $s \in \mathcal{S}$

$$g^* - g_{\pi^N}(s) \le \|Q^N - \mathcal{T}(Q^N)\|_{\mathrm{sp}} \le \varepsilon$$

*with sample and time complexity* $O\big(\widetilde{L}\,|\mathcal{S}||\mathcal{A}|\|h^*\|_{\mathrm{sp}}^2/\varepsilon^2\big)$ *where* $\widetilde{L} = \ln\big(|\mathcal{S}||\mathcal{A}|\|h^*\|_{\mathrm{sp}}/(\varepsilon\delta)\big) \ln^4(\|h^*\|_{\mathrm{sp}}/\varepsilon)$.

*Proof.* Since $Q^0 = 0$ we have $\|Q^0 - Q^*\|_{\mathrm{sp}} = \|Q^*\|_{\mathrm{sp}} = \|r + \mathcal{P} \max h^*\|_{\mathrm{sp}} \le \|r\|_{\mathrm{sp}} + \|h^*\|_{\mathrm{sp}}$. The result then follows from Theorem 3.4 by noting that $(1 + \mu/\varepsilon) = O(\|h\|_{\mathrm{sp}}^2/\varepsilon)$ which transforms $\widehat{L}$ into the multiplicative factor $\widetilde{L}$. $\qquad\square$

Next we proceed to establish the approximation and complexity results in expectation. We begin by the following technical Lemma which provides a crude bound for the sample complexity of each cycle of SAVIA+.

**Lemma A.4.** *Let* $M_i = |\mathcal{S}||\mathcal{A}|\sum_{j=0}^{n_i} m_j$ *be the number of samples used in* SAVIA$(Q^0, n_i, \varepsilon, \delta_i)$ *during a given cycle* $i \in \mathbb{N}$. *Then* $M_i \le |\mathcal{S}||\mathcal{A}|\,O\big(n_i + (\kappa/\varepsilon)^2 \alpha_i\,n_i^2 \ln^2(n_i + 2)\big)$ *where* $\alpha_i = \ln(|\mathcal{S}||\mathcal{A}|(n_i + 1)/\delta_i)$.

*Proof.* We first show by induction that for all $k = 0, \dots, n_i$ we have $\|d^k\|_{\mathrm{sp}} \le \kappa$ and $\|T^{k-1}\|_{\mathrm{sp}} \le (k + 1)\kappa$. For $k = 0$, this is true by initialization since $d^0 = \max_{\mathcal{A}} Q^0$ and $T^{-1} = r$. Using (6), (9), and the induction hypothesis, we have

$$\begin{aligned}
\|d^k\|_{\mathrm{sp}} \le \|Q^k - Q^{k-1}\|_{\mathrm{sp}} &\le \tfrac{2}{(k+1)(k+2)}\|T^{k-1} - Q^0\|_{\mathrm{sp}} + \tfrac{k-1}{k+1}\|d^{k-1}\|_{\mathrm{sp}} \\
&\le \tfrac{2}{(k+1)(k+2)}\big((k+1)\kappa + \kappa\big) + \tfrac{k-1}{k+1}\kappa = \kappa
\end{aligned}$$

and consequently

$$\|T^k\|_{\mathrm{sp}} \le \|T^{k-1}\|_{\mathrm{sp}} + \|D^k\|_{\mathrm{sp}} \le (k + 1)\kappa + \|d^k\|_{\mathrm{sp}} \le (k + 2)\kappa.$$

Then, using these bounds, the sample complexity $M_i$ for the $i$-th cycle can be readily bounded as

$$M_i \leq |\mathcal{S}||\mathcal{A}|((n_i + 1) + (\alpha_i/\varepsilon^2) \sum_{j=0}^{n_i} c_j \|d^j\|_{\mathrm{sp}}^2)$$
$$\leq |\mathcal{S}||\mathcal{A}|((n_i + 1) + 5(\kappa/\varepsilon)^2 \alpha_i \sum_{j=0}^{n_i} (j + 2) \ln^2(j + 2))$$
$$= |\mathcal{S}||\mathcal{A}| \, O\big(n_i + (\kappa/\varepsilon)^2 \alpha_i \, n_i^2 \ln^2(n_i + 2)\big).$$
$\square$

**Theorem 3.6.** *Assume* (H) *and* (S). *Let* $(Q^N, T^N, \pi^N)$ *be the output of* SAVIA+$(Q^0, \varepsilon, \delta)$. *Then, for all* $s \in \mathcal{S}$

$$\mathbb{E}[g^* - g_{\pi^N}(s)] \leq 16\,\varepsilon + \delta\|r\|_{\mathrm{sp}},$$

*with expected sample and time complexity*

$$\widetilde{O}\big(|\mathcal{S}||\mathcal{A}|(\nu^2/\varepsilon^2 + 1 + \delta\,(1 + \mu/\varepsilon)^2(1 + (\kappa/\varepsilon)^2)\big).$$

*Proof.* Consider the events $A = \{I \leq i_0\}$ and $B = \bigcap_{i=1}^{\infty} G_i$ as in the proof of Theorem 3.4 which established that on the event $A \cap B$ we have $g^* - g_{\pi^n}(s) \leq 16\,\varepsilon$ for all $s \in \mathcal{S}$ with $\mathbb{P}(A \cap B) \geq 1 - \delta$. On the complementary event $(A \cap B)^c$ we can use the crude bound $g^* - g_{\pi^n}(s) \leq \|r\|_{\mathrm{sp}}$ and then, since $\mathbb{P}((A \cap B)^c) \leq \delta$, we derive the first claim

$$\mathbb{E}[g^* - g_{\pi^n}(s)] \leq 16\,\varepsilon + \delta\|r\|_{\mathrm{sp}}.$$

To estimate the expected value of the sample complexity $M = \sum_{i=0}^{I} M_i$ we note that $A^c = \cup_{i=i_0+1}^{\infty}\{I = i\}$, and therefore

$$\mathbb{E}[M] = \mathbb{E}[M|A \cap B]\,\mathbb{P}(A \cap B) + \mathbb{E}[M|A \cap B^c]\,\mathbb{P}(A \cap B^c) + \sum_{i=i_0+1}^{\infty} \mathbb{E}[M|I = i]\,\mathbb{P}(I = i).$$

Let us bound separately the three terms in this sum. For the first term we observe that on the event $A \cap B$ we can apply the bound in Theorem 3.4 and since $\mathbb{P}(A \cap B) \leq 1$ we get with $\widehat{L} = \ln\big(|\mathcal{S}||\mathcal{A}|\,(1+\mu/\varepsilon)/\delta\big)\,\ln^4(1+\mu/\varepsilon)$ that

$$\mathbb{E}[M|A \cap B]\,\mathbb{P}(A \cap B) = O\big(\widehat{L}|\mathcal{S}||\mathcal{A}|(\nu^2/\varepsilon^2 + 1)\big).$$

For the second term, we combine $\mathbb{P}(A \cap B^c) \leq \mathbb{P}(B^c) \leq \delta$ with the bound in Lemma A.4 and $n_{i_0} \leq 2(1 + \mu/\varepsilon)$ to get

$$\mathbb{E}[M|A \cap B^c]\,\mathbb{P}(A \cap B^c) \leq \delta\,|\mathcal{S}||\mathcal{A}| \sum_{i=0}^{i_0} O\big(n_i + (\kappa/\varepsilon)^2 \alpha_i n_i^2 \ln^2(n_i + 2)\big)$$
$$\leq \delta\,|\mathcal{S}||\mathcal{A}|\,O\big(n_{i_0} + (\kappa/\varepsilon)^2 \alpha_{i_0} n_{i_0}^2 \ln^3(n_{i_0} + 2)\big)$$
$$\leq \delta\,|\mathcal{S}||\mathcal{A}|\,O\big(n_{i_0} + \widehat{L}\,(\kappa/\varepsilon)^2 n_{i_0}^2\big)$$

where the last inequality follows from the bound $\alpha_{i_0} \ln^3(n_{i_0} + 2) \leq O(\widehat{L})$.

For the third term $R \triangleq \sum_{i=i_0+1}^{\infty} \mathbb{E}[M|I = i]\,\mathbb{P}(I = i)$, using again the bound in Lemma A.4 we get

$$\mathbb{E}[M|I = i] = \sum_{j=0}^{i} \mathbb{E}[M_j] \leq \sum_{j=0}^{i} |\mathcal{S}||\mathcal{A}|\,O\big(n_j + (\kappa/\varepsilon)^2 \alpha_j n_j^2 \ln^2(n_j + 2)\big)$$
$$= |\mathcal{S}||\mathcal{A}|\,O\big(n_i + (\kappa/\varepsilon)^2 \alpha_i n_i^2 \ln^2(n_i + 2)\big)$$

and therefore

$$R \leq |\mathcal{S}||\mathcal{A}| \sum_{i=i_0+1}^{\infty} O\big(n_i + (\kappa/\varepsilon)^2 \alpha_i n_i^2 \ln^2(n_i + 2)\big)\,\mathbb{P}(I = i).$$

To estimate the dominant terms in this last sum we recall that $\alpha_i = \ln(2|\mathcal{S}||\mathcal{A}|(n_i + 1)/\delta_i)$ with $n_i = 2^i$ and $\delta_i = \delta/c_i$. Since $\ln((n_i + 1)c_i) = O(i)$ we get $\alpha_i = O(\widehat{L} + i) \leq \widehat{L}\,O(i)$, and similarly $\ln^2(n_i + 2) = O(i^2)$, so that

$$R \leq |\mathcal{S}||\mathcal{A}| \sum_{i=i_0+1}^{\infty} O\big(n_i + \widehat{L}\,(\kappa/\varepsilon)^2\,i^3 n_i^2\big)\,\mathbb{P}(I = i).$$

Using the fact that $n_i = n_{i_0} 2^{i-i_0}$ and noting that for $i \geq i_0 + 1$ we have $\mathbb{P}(I = i) \leq \prod_{j=i_0}^{i-1} \delta_j \leq O\big(\delta \prod_{j=i_0}^{i-1} \frac{1}{j+2}\big)$ (see the proof of Proposition 3.3), by setting $S_1 = \sum_{i=i_0+1}^{\infty} 2^{i-i_0} \prod_{j=i_0}^{i-1} \frac{1}{j+2}$ and $S_2 = \sum_{i=i_0+1}^{\infty} 2^{2(i-i_0)} i^3 \prod_{j=i_0}^{i-1} \frac{1}{j+2}$ we derive

$$R \leq \delta\,|\mathcal{S}||\mathcal{A}|\,O\big(S_1\,n_{i_0} + S_2\,\widehat{L}\,(\kappa/\varepsilon)^2 n_{i_0}^2\big).$$

The sums $S_1$ and $S_2$ can be computed explicitly in terms of incomplete Gamma functions as

$$S_1 = e^2 \, 2^{-(i_0+1)}[\Gamma(i_0+2) - \Gamma(i_0+2,2)] \leq (e^2-3)/2,$$
$$S_2 = 84 + 4i_0(i_0+5) + 67e^4 2^{-2(i_0+1)}[\Gamma(i_0+2) - \Gamma(i_0+2,4)] = O((i_0+1)^2).$$

so the third term satisfies

$$R \leq \delta \, |\mathcal{S}||\mathcal{A}| \, O\Big(n_{i_0} + \widehat{L}\,(\kappa/\varepsilon)^2 n_{i_0}^2 (i_0+1)^2\Big).$$

Putting together these three bounds, by regrouping terms of the same order and ignoring logarithmic terms we get the announced bound for the expected value of the sample complexity

$$\mathbb{E}[M] \leq |\mathcal{S}||\mathcal{A}| \, O\left(\widehat{L}(\nu^2/\varepsilon^2 + 1) + \delta \, n_{i_0} + \delta \, \widehat{L}\,(\kappa/\varepsilon)^2 n_{i_0}^2 (i_0+1)^2\right)$$
$$= |\mathcal{S}||\mathcal{A}| \, \widetilde{O}\left((\nu^2/\varepsilon^2 + 1) + \delta \,(1 + \mu/\varepsilon)^2 (1 + (\kappa/\varepsilon)^2)\right).$$

$\square$

**Corollary 3.7.** *Assume* (H), (S), $r(s,a) \in [0,1]$ *for all* $(s,a) \in \mathcal{S} \times \mathcal{A}$ *and* $\|h^*\|_{\mathrm{sp}} \geq 1$. *Let* $(Q^N, T^N, \pi^N)$ *be the output of* SAVIA+$(Q^0, \varepsilon/17, \delta)$ *with* $Q^0 = 0$, $\varepsilon \leq 1$, *and* $\delta = \varepsilon^2/17$. *Then, for all* $s \in \mathcal{S}$

$$\mathbb{E}[g^* - g_{\pi^N}(s)] \leq \varepsilon,$$

*with expected sample complexity* $\widetilde{O}\left(|\mathcal{S}||\mathcal{A}|\|h^*\|_{\mathrm{sp}}^2/\varepsilon^2\right)$.

*Proof.* From Theorem 3.6 we have $\mathbb{E}[g^* - g_{\pi^n}(s)] \leq \frac{16}{17}\varepsilon + \frac{\varepsilon^2}{17}\|r\|_{\mathrm{sp}} \leq \varepsilon$. Also the expected complexity follows directly from Theorem 3.6 and the choice of $\delta$. $\square$

# B. Omitted proofs for the discounted reward setting

A nice feature of our approach, presented mainly in the average reward case, is that it can be applied almost verbatim to the discounted setup. Consequently, the proofs for discounted MDPs and average reward MDPs are fairly similar. However, for the sake of completeness, the arguments are repeated or referred to those in Appendix A when they are basically the same.

## B.1. Proofs of Section 4.2

**Proposition 4.1.** *Let* $Q \in \mathbb{R}^{\mathcal{S} \times \mathcal{A}}$ *and* $\pi : \mathcal{S} \to \mathcal{A}$ *a greedy policy such that* $\pi(s) \in \operatorname{argmax}_{a \in \mathcal{A}} Q(s,a)$. *Then we have* $\|Q^* - Q_\pi\|_\infty \leq \frac{2}{1-\gamma} \|Q - \mathcal{T}_\gamma(Q)\|_\infty$.

*Proof.* A simple triangle inequality gives $\|Q^* - Q_\pi\|_\infty \leq \|Q^* - Q\|_\infty + \|Q_\pi - Q\|_\infty$ so it suffices to show that both terms in this latter sum are bounded by $\frac{1}{1-\gamma}\|Q - \mathcal{T}_\gamma(Q)\|_\infty$. The first inequality $\|Q^* - Q\|_\infty \leq \frac{1}{1-\gamma}\|Q - \mathcal{T}_\gamma(Q)\|_\infty$ results directly from the following triangle inequality combined with $Q^* = \mathcal{T}_\gamma(Q^*)$ and the fact that $\mathcal{T}_\gamma$ is a $\gamma$-contraction

$$\|Q^* - Q\|_\infty \leq \|Q^* - \mathcal{T}_\gamma(Q)\|_\infty + \|\mathcal{T}_\gamma(Q) - Q\|_\infty \leq \gamma\|Q^* - Q\|_\infty + \|Q - \mathcal{T}_\gamma(Q)\|_\infty.$$

To establish the second inequality $\|Q_\pi - Q\|_\infty \leq \frac{1}{1-\gamma}\|Q - \mathcal{T}_\gamma(Q)\|_\infty$, we first note that $Q_\pi$ satisfies the recursive formula

$$Q_\pi(s,a) = r(s,a) + \gamma\sum_{s' \in \mathcal{S}} p(s'|s,a)\, Q_\pi(s', \pi(s'))$$

while the definition of $\pi$ gives $Q(s', \pi(s')) = \max_\mathcal{A}(Q)(s')$ so that

$$\mathcal{T}_\gamma(Q)(s,a) = r(s,a) + \gamma\sum_{s' \in \mathcal{S}} p(s'|s,a)\, Q(s', \pi(s')).$$

Then, substracting both formulas we readily get $\|Q_\pi - \mathcal{T}_\gamma(Q)\|_\infty \leq \gamma\|Q_\pi - Q\|_\infty$ and therefore

$$\|Q_\pi - Q\|_\infty \leq \|Q_\pi - \mathcal{T}_\gamma(Q)\|_\infty + \|\mathcal{T}_\gamma(Q) - Q\|_\infty \leq \gamma\|Q_\pi - Q\|_\infty + \|\mathcal{T}_\gamma(Q) - Q\|_\infty$$

which readily implies $\|Q_\pi - Q\|_\infty \leq \frac{1}{1-\gamma}\|Q - \mathcal{T}_\gamma(Q)\|_\infty$ as claimed. $\square$

Analogously to the average reward setting, let $\mathcal{F}_k = \sigma(\{D^i : i = 0, \ldots, k\})$ be the natural filtration generated by the sampling process in SAVID, and $\mathbb{P}(\cdot)$ the probability distribution over the trajectories $(D^k)_{k \in \mathbb{N}}$. Again, $T^k$ is $\mathcal{F}_k$-measurable whereas $Q^k$, $V^k$, $d^k$, and $m_k$ are $\mathcal{F}_{k-1}$-measurable.

**Proposition B.1.** *Let $c_k > 0$ with $2 \sum_{k=0}^{\infty} c_k^{-1} \leq 1$ and $T^k, Q^k$ the iterates generated by* SAVID($Q^0, n, \varepsilon, \delta, \gamma$). *Then, with probability at least $1 - \delta$ we have $\|T^k - \mathcal{T}_\gamma(Q^k)\|_\infty \leq \gamma \varepsilon$ simultaneously for all $k = 0, \ldots, n$.*

*Proof.* Let $Y^i = D^i - \mathcal{P}d^i$ and $X^k = \sum_{i=0}^{k} Y^i$. Since $V^{-1} = 0$, by telescoping $\mathcal{T}_\gamma(Q^k) = r + \gamma \mathcal{P}h^k = r + \gamma \sum_{i=0}^{k} \mathcal{P}d^i$ and then using (2) we get $T^k - \mathcal{T}_\gamma(Q^k) = \gamma \sum_{i=0}^{k}(D^i - \mathcal{P}d^i) = \gamma X^k$. The estimation of $\mathbb{P}(\|X^k\|_\infty \geq \varepsilon)$ proceeds in the exact same manner as in the proof of Proposition 3.1. $\square$

From now on we consider the specific sequences $c_k = 5(k+2)\ln^2(k+2)$ and $\beta_k = k/(k+2)$, meaning that we assume (S). Let $Q^*$ be the unique solution of the Bellman equation $\mathcal{T}_\gamma(Q) = Q$. We assume throughout that

$$\left\|T^k - \mathcal{T}_\gamma(Q^k)\right\|_\infty \leq \varepsilon \qquad \text{for all } k = 0, \ldots, n. \tag{12}$$

which, in view of Proposition B.1, holds with probability at least $(1 - \delta)$ when SAVID($Q^0, n, \varepsilon, \delta, \gamma$) is implemented.

Also, as noted earlier, for $D = \text{SAMPLE}(d, m)$ we have $\|D\|_\infty \leq \|d\|_\infty$, which combined with the nonexpansivity of the map $Q \mapsto \max_{\mathcal{A}}(Q)$ implies

$$\left\|T^k - T^{k-1}\right\|_\infty = \gamma \left\|D^k\right\|_\infty \leq \gamma \left\|d^k\right\|_\infty = \gamma \left\|V^k - V^{k-1}\right\|_\infty \leq \gamma \left\|Q^k - Q^{k-1}\right\|_\infty. \tag{13}$$

The following two technical lemmas serve as counterparts to Lemmas A.1 and A.2 in this setting. Unlike the previous results, they establish estimates from the iterate $Q^k$ to $Q^*$ in the infinity norm and therefore the conclusions change slightly. As before, these lemmas act as a prelude to the general bound given by Theorem B.4.

**Lemma B.2.** *Assuming* (S) *and* (12), *we have that*

$$\left\|Q^k - Q^*\right\|_\infty \leq \left\|Q^0 - Q^*\right\|_\infty + \frac{1}{3}\varepsilon k \quad \text{for all } k = 0, \ldots, n.$$

*Proof.* From the iteration $Q^k = (1 - \beta_k)Q^0 + \beta_k T^{k-1}$ with $\beta_k = \frac{k}{k+2}$ we get

$$\left\|Q^k - Q^*\right\|_\infty \leq \tfrac{2}{k+2}\left\|Q^0 - Q^*\right\|_\infty + \tfrac{k}{k+2}\left\|T^{k-1} - Q^*\right\|_\infty.$$

Bound (12) and $\gamma \leq 1$ imply

$$\left\|T^{k-1} - Q^*\right\|_\infty = \left\|T^{k-1} - \mathcal{T}_\gamma(Q^*)\right\|_\infty \leq \varepsilon + \gamma \left\|Q^{k-1} - Q^*\right\|_\infty \leq \varepsilon + \left\|Q^{k-1} - Q^*\right\|_\infty. \tag{14}$$

From here the proof follows the exact same lines as in the proof of Lemma A.1. $\square$

**Lemma B.3.** *Assume* (S) *and* (12) *and let $\rho_k = 2\left\|Q^0 - Q^*\right\|_\infty + \frac{1}{3}\varepsilon k$. Then*

$$\left\|Q^k - Q^{k-1}\right\|_\infty \leq \frac{2}{k(k+1)} \sum_{i=1}^{k} \rho_{i+2}. \qquad \text{for all } k = 1, \ldots, n.$$

*Proof.* From $Q^k = \frac{2}{k+2}Q^0 + \frac{k}{k+2}T^{k-1}$ and $Q^{k-1} = \frac{2}{k+1}Q^0 + \frac{k-1}{k+1}T^{k-2}$ we derive

$$Q^k - Q^{k-1} = \tfrac{2}{(k+1)(k+2)}(T^{k-1} - Q^0) + \tfrac{k-1}{k+1}(T^{k-1} - T^{k-2}).$$

Now, (14) together with Lemma B.2 readily imply $\left\|T^{k-1} - Q^0\right\|_\infty \leq \left\|T^{k-1} - Q^*\right\|_\infty + \left\|Q^* - Q^0\right\|_\infty \leq \rho_{k+2}$, and therefore using (13) we get

$$\left\|Q^k - Q^{k-1}\right\|_\infty \leq \tfrac{2}{(k+1)(k+2)}\rho_{k+2} + \gamma \tfrac{k-1}{k+1}\left\|Q^{k-1} - Q^{k-2}\right\|_\infty.$$

Using that $\gamma \leq 1$, the proof finishes exactly as in Lemma A.2 $\square$

The following theorem is the analogue of Theorem 3.2 for discounted MDPs. For the sake of completeness and given that the constants involved are different, we provide the full proof.

**Theorem B.4.** *Assume* (S) *and let* $(Q^n, T^n, \pi^n)$ *be the output computed by* SAVID$(Q^0, n, \varepsilon, \delta, \gamma)$. *Then, with probability at least* $(1 - \delta)$ *we have*

$$\|Q^n - \mathcal{T}_\gamma(Q^n)\|_\infty \leq \frac{8\|Q^0 - Q^*\|_\infty}{n+2} + 2\varepsilon,$$

*with a sample and time complexity of order*

$$O\big(L_\gamma |\mathcal{S}||\mathcal{A}|((\|Q^0 - Q^*\|_\infty^2 + \|Q^0\|_\infty^2)/\varepsilon^2 + n^2)\big)$$

*where* $L_\gamma = \ln(2|\mathcal{S}||\mathcal{A}|(n+1)/\delta)\ln^3(n+2)$.

*Proof.* From Proposition B.1, with probability at least $(1 - \delta)$ we have $\|T^k - \mathcal{T}_\gamma(Q^k)\|_\infty \leq \varepsilon$ for all $k = 0, \ldots, n$. Now, the recursion $Q^n = \frac{2}{n+2}Q^0 + \frac{n}{n+2}T^{n-1}$ implies

$$Q^n - \mathcal{T}_\gamma(Q^n) = \frac{2}{n+2}\big(Q^0 - \mathcal{T}_\gamma(Q^n)\big) + \frac{n}{n+2}\big(T^{n-1} - \mathcal{T}_\gamma(Q^{n-1})\big) + \frac{n}{n+2}\big(\mathcal{T}_\gamma(Q^{n-1}) - \mathcal{T}_\gamma(Q^n)\big). \tag{15}$$

Using the triangle inequality and Lemma B.2, we have that

$$\|Q^0 - \mathcal{T}_\gamma(Q^n)\|_\infty \leq \|Q^0 - Q^*\|_\infty + \gamma\|Q^* - Q^n\|_\infty \leq 2\|Q^0 - Q^*\|_\infty + \tfrac{1}{3}\varepsilon n = \rho_n,$$

while Lemma B.3 gives $\|\mathcal{T}_\gamma(Q^{n-1}) - \mathcal{T}_\gamma(Q^n)\|_\infty \leq \frac{2}{n(n+1)}\sum_{i=1}^n \rho_{i+2}$. Thus, applying a triangle inequality to (15) and using these estimates together with Proposition 4.1, we obtain

$$
\begin{aligned}
\|Q^n - \mathcal{T}_\gamma(Q^n)\|_\infty &\leq \tfrac{2}{n+2}\rho_n + \varepsilon + \tfrac{2}{(n+1)(n+2)}\sum_{i=1}^n \rho_{i+2} \\
&= \tfrac{4(1+2n)}{(n+1)(n+2)}\|Q^0 - Q^*\|_\infty + \tfrac{2(3+8n+3n^2)}{3(n+1)(n+2)}\varepsilon \\
&\leq \tfrac{8\|Q^0 - Q^*\|_\infty}{n+2} + 2\varepsilon.
\end{aligned}
$$

To estimate the complexity, for $k \geq 1$, we use the inequality $\|d^k\|_\infty \leq \|Q^k - Q^{k-1}\|_\infty$ in (13) together with Lemma B.3 to find

$$
\begin{aligned}
\|d^k\|_\infty &\leq \tfrac{2}{k(k+1)}\sum_{i=1}^k \rho_{i+2} \\
&= \tfrac{4}{k+1}\|Q^0 - Q^*\|_\infty + \tfrac{k+5}{3(k+1)}\varepsilon \\
&\leq \tfrac{4}{k+1}\|Q^0 - Q^*\|_\infty + \varepsilon.
\end{aligned}
$$

Now, to estimate the total number of samples $|\mathcal{S}||\mathcal{A}|\sum_{k=0}^n m_k$ we recall that $m_k = \max\{\lceil 2\alpha c_k \|d^k\|_\infty^2/\varepsilon^2\rceil, 1\}$ which can be bounded as $m_k \leq 1 + 2\alpha c_k \|d^k\|_\infty^2/\varepsilon^2$. Then

$$
\begin{aligned}
\sum_{k=0}^n m_k &\leq (n+1) + \tfrac{2\alpha}{\varepsilon^2}\sum_{k=0}^n c_k \|d^k\|_\infty^2 \\
&\leq (n+1) + \tfrac{20\alpha}{\varepsilon^2}\ln^2(2)\|Q_0\|_\infty^2 + \tfrac{10\alpha}{\varepsilon^2}\sum_{k=1}^n (k+2)\ln^2(k+2)\big(\tfrac{4}{k+1}\|Q^0 - Q^*\|_\infty + \varepsilon\big)^2 \\
&\leq (n+1) + \tfrac{20\alpha}{\varepsilon^2}\ln^2(2)\|Q_0\|_\infty^2 + \sum_{k=1}^n \tfrac{480\alpha}{\varepsilon^2(k+2)}\ln^2(k+2)\|Q^0 - Q^*\|_\infty^2 + 20\alpha\sum_{k=1}^n (k+2)\ln^2(k+2) \\
&= O\big(\alpha\|Q^0\|_\infty^2/\varepsilon^2 + \alpha\ln^3(n+2)\|Q^0 - Q^*\|_\infty^2/\varepsilon^2 + \alpha n^2\ln^2(n+2)\big), \tag{16}
\end{aligned}
$$

where we use the same estimations as in the proof of Theorem 3.2. □

**Theorem 4.2.** *Assume* (S), $r(s,a) \in [0,1]$ *for all* $(s,a) \in \mathcal{S} \times \mathcal{A}$, *and* $n = \lceil 10/((1-\gamma)\varepsilon)\rceil$. *Let* $(Q^n, T^n, \pi^n)$ *be the output of* SAVID$(Q^0, n, \epsilon/10, \delta, \gamma)$ *with* $Q^0 = 0$ *and* $\varepsilon \leq 1/(1-\gamma)$. *Then, with probability at least* $(1 - \delta)$ *we have*

$$\|Q^n - \mathcal{T}_\gamma(Q^n)\|_\infty \leq \varepsilon$$

*with sample and time complexity* $O\big(L_\gamma|\mathcal{S}||\mathcal{A}|/((1-\gamma)^2\varepsilon^2)\big)$ *where* $L_\gamma = \ln(2|\mathcal{S}||\mathcal{A}|/((1-\gamma)\varepsilon\delta))\ln^3(2/((1-\gamma)\varepsilon))$.

*Proof.* The proof follows directly from Theorem B.4 and the bound $\|Q^*\|_\infty \leq 1/(1-\gamma)$. □

## B.2. Proofs of Section 4.3

We now proceed to establish the complexity of the algorithm SAVID+. Again, the proof is almost the same as in the average reward case, where some of the constants must be estimated differently. Precisely, the time in which the inner loop in SAVID+ stops will now depend on $\left\|Q^0 - Q^*\right\|_\infty$.

Under a slight abuse of notation, we define the stopping time of SAVID+ as

$$N = \inf\{n_i \in \mathbb{N} \,:\, \|Q^{n_i} - T^{n_i}\|_\infty \le 11\,\varepsilon\},$$

where $T^{n_i}$ and $Q^{n_i}$'s are the iterates generated in each loop of SAVID+$(Q^0, \varepsilon, \delta, \gamma)$. As before, we let $i_0 \in \mathbb{N}$ be the smallest integer satisfying $n_{i_0} \ge \left\|Q^0 - Q^*\right\|_\infty / \varepsilon$, so that either $i_0 = 0$ and $n_{i_0} = 1$ or $n_{i_0-1} = n_{i_0}/2 < \left\|Q^0 - Q^*\right\|_\infty / \varepsilon$, which combined imply $n_{i_0} \le 1 + 2\left\|Q^0 - Q^*\right\|_\infty / \varepsilon$. Also, as in the average reward setting, we consider the events

$$S_i = \{\|Q^{n_i} - T^{n_i}\|_\infty \le 11\,\varepsilon\} \quad\text{and}\quad G_i = \{\left\|T^k - \mathcal{T}_\gamma(Q^k)\right\|_\infty \le \varepsilon,\ \forall k = 0, \dots, n_i\},$$

with $Q^k$ and $T^k$'s the inner iterates generated during the execution of SAVID$(Q^0, n_i, \varepsilon, \delta_i, \gamma)$ in the $i$-th loop of SAVID+.

**Lemma B.5.** *Assume* (S). *Then, for all $i \ge i_0$ we have that $\mathbb{P}(S_i) \ge \mathbb{P}(G_i) \ge 1 - \delta_i$.*

*Proof.* The proof is the same as its average reward counterpart Lemma A.3 by just observing that, if $i \ge i_0$ then, for all $\omega \in G_i$,

$$\|Q^{n_i}(\omega) - T^{n_i}(\omega)\|_\infty \le \|Q^{n_i}(\omega) - \mathcal{T}_\gamma(Q^{n_i})(\omega)\|_\infty + \|\mathcal{T}_\gamma(Q^{n_i})(\omega) - T^{n_i}(\omega)\|_\infty$$

$$\le \frac{8\|Q^0 - Q^*\|_\infty}{n_i + 2} + 2\,\varepsilon + \varepsilon$$

$$\le 11\,\varepsilon. \qquad\qquad\square$$

**Theorem B.6.** *Assume* (S) *and let $(Q^N, T^N, \pi^N)$ be the output of SAVID+$(Q^0, \varepsilon, \delta, \gamma)$. Then, with probability at least $(1 - \delta)$ we have*

$$\left\|Q^N - \mathcal{T}_\gamma(Q^N)\right\|_\infty \le 12\,\varepsilon$$

*with sample and time complexity*

$$O\left(\widehat{L}_\gamma |\mathcal{S}||\mathcal{A}| \left(\left(\left\|Q^0\right\|_\infty + \left\|Q^0 - Q^*\right\|_\infty\right)^2/\varepsilon^2 + 1\right)\right),$$

*where $\widehat{L}_\gamma = \ln\big(|\mathcal{S}||\mathcal{A}|\,(1 + 2\left\|Q^0 - Q^*\right\|_\infty/\varepsilon)/\delta\big)\,\ln^4(1 + 2\left\|Q^0 - Q^*\right\|_\infty/\varepsilon)$.*

*Proof.* Consider the events $A = \{I \le i_0\}$ and $B = \bigcap_{i=0}^{\infty} G_i$. Using the exact same argument as in the proof of Theorem 3.4, now through Proposition B.1 and Lemma B.5, we know that $\mathbb{P}(A \cap B) \ge 1 - \delta$. Also, on the event $A \cap B$ and from the definition of $N$, we have

$$\left\|Q^N - \mathcal{T}_\gamma(Q^N)\right\|_\infty \le \left\|Q^N - T^N\right\|_\infty + \left\|T^N - \mathcal{T}_\gamma(Q^N)\right\|_\infty \le 11\,\varepsilon + \varepsilon = 12\,\varepsilon.$$

Now, using (16) in the proof of Theorem B.4 the total sample complexity $M$ of the algorithm can be estimated, as in the average reward case, by

$$M \le |\mathcal{S}||\mathcal{A}| \sum_{i=0}^{i_0} O\left(\alpha_i \left\|Q^0\right\|_\infty^2/\varepsilon^2 + \alpha_i \ln^3(n_i + 2)\left\|Q^0 - Q^*\right\|_\infty^2/\varepsilon^2 + \alpha_i\, n_i^2 \ln^2(n_i + 2)\right),$$

where $\alpha_i = \ln(2|\mathcal{S}||\mathcal{A}|(n_i + 1)/\delta_i)$ is the parameter defined in the $i$-th cycle of SAVID+. Again, as for SAVIA+, we use that $n_{i_0}^2 \le (1 + 2\left\|Q^0 - Q^*\right\|_\infty/\varepsilon)^2 = O(\left\|Q^0 - Q^*\right\|_\infty^2/\varepsilon^2 + 1)$, to get

$$M \le |\mathcal{S}||\mathcal{A}|\, O\left(\alpha_{i_0} \left\|Q^0\right\|_\infty^2/\varepsilon^2 \ln(n_{i_0} + 2) + \alpha_{i_0} \ln^4(n_{i_0} + 2)\left\|Q^0 - Q^*\right\|_\infty^2/\varepsilon^2 + \alpha_{i_0}\, n_{i_0}^2 \ln^4(n_{i_0} + 2)\right)$$

$$\le |\mathcal{S}||\mathcal{A}|\,\alpha_{i_0} \ln^4(n_{i_0} + 2)\, O\left(\left(\left\|Q^0\right\|_\infty + \left\|Q^0 - Q^*\right\|_\infty\right)^2/\varepsilon^2 + 1\right).$$

$$\qquad\qquad\square$$

Finally, we state our claimed complexity result for SAVID**+**, whose proof follows directly from Proposition B.1, Theorem B.6, and the fact that $\|Q^*\|_\infty \le 1/(1-\gamma)$.

**Theorem 4.3.** *Assume* (S)*,* $r(s, a) \in [0, 1]$ *for all* $(s, a) \in \mathcal{S} \times \mathcal{A}$*, and* $\|Q^*\|_\infty \ge 1$*. Let* $(Q^N, T^N, \pi^N)$ *be the output of* SAVID**+**$(Q^0, \varepsilon(1-\gamma)/24, \delta, \gamma)$ *with* $Q^0 = 0$ *and* $\varepsilon \le 1/(1-\gamma)$*. Then, with probability at least* $(1 - \delta)$ *we have*

$$\|Q^* - Q_{\pi^N}\|_\infty \le 2\|Q^N - \mathcal{T}_\gamma(Q^N)\|_\infty /(1-\gamma) \le \varepsilon$$

*with sample and time complexity*

$$O\left(\widetilde{L}_\gamma |\mathcal{S}||\mathcal{A}| \|Q^*\|_\infty^2 /((1-\gamma)^2 \varepsilon^2)\right)$$

*where* $\widetilde{L}_\gamma = \ln\left(2|\mathcal{S}||\mathcal{A}|/((1-\gamma)\varepsilon\delta)\right) \ln^4\left(2/((1-\gamma)\varepsilon)\right)$*.*

## C. The Anchored Value Iteration as Halpern's iteration on a quotient space

The recursion (1) involves the unknown optimal value $g^*$. However, since $\mathcal{T}(Q + c) = \mathcal{T}(Q) + c$ is homogeneous under addition of constants $c \in \mathbb{R}$, this prompts us to consider the map $\widetilde{\mathcal{T}} : \mathcal{M} \to \mathcal{M}$ defined by $\widetilde{\mathcal{T}}([Q]) = [\mathcal{T}(Q)]$ on the quotient space $\mathcal{M} = \mathbb{R}^{\mathcal{S} \times \mathcal{A}}/E$ with $E$ the subspace of constant matrices, endowed with the induced quotient norm

$$\big\|[Q]\big\|_E = \min_{c \in \mathbb{R}} \big\|Q + c\big\|_\infty = \frac{1}{2}\|Q\|_{\mathrm{sp}}.$$

One can readily check that $\widetilde{\mathcal{T}}$ is nonexpansive for the norm $\|\cdot\|_E$ and moreover

CLAIM: $Q^* \in \mathrm{Fix}(\mathcal{T}_{g^*})$ if and only if $[Q^*] \in \mathrm{Fix}(\widetilde{\mathcal{T}})$.

*Proof.* If $Q^* = \mathcal{T}_{g^*}(Q^*)$ then $[Q^*] = [\mathcal{T}(Q^* - g^*)] = \widetilde{\mathcal{T}}([Q^* - g^*]) = \widetilde{\mathcal{T}}([Q^*])$. Conversely, if $[Q^*] = \widetilde{\mathcal{T}}([Q^*]) = [\mathcal{T}(Q^*)]$ then $Q^* = \mathcal{T}(Q^*) + c$ for some $c \in \mathbb{R}$, and Puterman (2014, Theorem 9.1.2) gives $c = -g^*$ so that $Q^* \in \mathrm{Fix}(\mathcal{T}_{g^*})$. $\square$

Projecting (1) on the quotient space $\mathcal{M}$ it follows that the equivalence classes $[Q^k] = Q^k + E$ of the iterates $\{Q^k\}_{k \in \mathbb{N}}$ generated by (1) satisfy the following Halpern's iteration for $\widetilde{\mathcal{T}}(\cdot)$

$$[Q^{k+1}] = (1 - \beta_{k+1})[Q^0] + \beta_{k+1}\widetilde{\mathcal{T}}([Q^k]).$$

In this projected iteration the unknown $g^*$ plays no role. Moreover, the equivalence classes $[Q^n]$ coincide with those generated by the implementable modification (Anc-VI) in which $g^*$ is ignored. This shows that, modulo constants, both (1) and (Anc-VI) are equivalent and their corresponding residuals in span seminorm coincide.

Finally, using the identity $\|Q - \mathcal{T}(Q)\|_{\mathrm{sp}} = 2\big\|[Q] - \widetilde{\mathcal{T}}([Q])\big\|$, any error bound for Halpern's iteration as applied to $\widetilde{\mathcal{T}}$ directly transfers into a bound for $\|Q^k - \mathcal{T}(Q^k)\|_{\mathrm{sp}}$. In particular, for $\beta_k = k/(k + 2)$ we get

$$\|Q^k - \mathcal{T}(Q^k)\|_{\mathrm{sp}} \le \frac{4}{k+1}\|Q^0 - Q^*\|_{\mathrm{sp}}.$$

