# OpenReview forum: "Near-Optimal Sample Complexity for MDPs via Anchoring"
_ICML.cc/2025/Conference — ICML 2025 poster_

### Official Review · Reviewer_hCuT · 2025-03-10

**Overall Recommendation:** 4

**Summary:**

The authors propose a new model-free algorithm for solving average-reward weakly communicating MDPs with a generative model. The authors achieved the sample complexity of order $\widetilde{O}(SAH^2 / \varepsilon^2)$, where $S$ is a number of states, $A$ is a number of actions, $H$ is a span of an optimal bias function. The main property of the algorithm is that it does not require any prior knowledge on $H$. Additionally, the authors adapted their approach to a discounted setting.

**Claims And Evidence:**

**Claim 1.** Algorithm SAVIA achieves the $\varepsilon$-policy error using $\mathcal{O}(SAH^2/\varepsilon^2)$ samples but requires the knowledge of $H$;

The proof looks good to me.

**Claim 2.** Algorithm SAVIA+ achieves the $\varepsilon$-policy error using $\mathcal{O}(SAH^2/\varepsilon^2)$ samples without knowledge of $H$, using a combination of SAVIA and the doubling trick.

The statement is expected and looks good to me too, however, I did not verify the proof in detail.

**Claim 3.** This algorithm also achieves $\varepsilon$-error in terms of solution to Bellman equations with the same sample complexity;

This automatically follows from the structure of the proof.

**Claim 4.** The adaptation of the algorithm to discounting case achieves $\varepsilon$-solution to Bellman equations using $\mathcal{O}(SA/(\varepsilon^2 (1-\gamma)^2))$ samples and $\varepsilon$-optimal policy using $\mathcal{O}(SA/(\varepsilon^2 (1-\gamma)^4))$ samples.

This statement looks good to me, although I did not verify it in detail in the Appendix.

**Essential References Not Discussed:**

All related literature was discussed in detail.

**Experimental Designs Or Analyses:**

N/A

**Methods And Evaluation Criteria:**

The paper of theoretical nature and does not contain any empirical studies.

**Other Comments Or Suggestions:**

No additional comments.

**Other Strengths And Weaknesses:**

As one of the strengths of the paper, I have to underline the clarity and the quality of writing, as well as a detailed literature review.

**Questions For Authors:**

- What do you think, could a variance-reduction technique (Wainwright, 2019) reduce the sample complexity of your method in the average-reward setting? For example, it is observable that in the current approach, the dependence on H^2 comes from evaluation of expectations with respect to $d^k$, but if one replaces $d^k$ with some variance-reduced version with a norm of only a constant order, it can improve the final sample complexity.
- How is the proposed method connected to a usual Q-learning? Can we treat this method as some modification of Q-learning?


Wainwright, M. J. (2019). Variance-reduced $ Q $-learning is minimax optimal. arXiv preprint arXiv:1906.04697.

**Relation To Broader Scientific Literature:**

The key contributions of this paper follow the current scientific literature. In particular, the question on the optimal sample complexity without prior knowledge in average-reward MDPs is very interesting and not yet well-studied (although the number of studies of this question increases).

**Theoretical Claims:**

See **Claims and Evidence** section.

---

> ### Author Rebuttal · Authors · 2025-04-01
>
> Thank you for the thoughtful comments. Both of the Reviewer's questions are central to our motivation for writing this work. For the list of references please refer to the reply to Reviewer BWFD.
>
> 1. Use of a related variance reduction technique
>
> The paper [16] is related to ours in the sense that classical (synchronous) Q-learning is fundamentally a stochastic version of the Krasnoselskii-Mann (KM) iteration for a contracting operator. It builds on earlier work by the same author [17], where a sample complexity of $O(SA(1-\gamma)^{-5}\epsilon^{-2})$ was established. However, it is important to note that the method cited by the Reviewer applies only in the discounted setting, with complexity measured in terms of the distance to the optimal Q-factor. The argument in that paper is clever: by employing variance reduction techniques inspired by stochastic optimization, it improves the original (suboptimal) complexity by a factor of $1/(1-\gamma)$. To achieve minimax optimality, a rerun of the main algorithm is carefully designed. From a technical standpoint, the variance reduction argument relies on a precise interplay between the step size (which depends on the contraction parameter), the number of iteration $n$, and the specific structure of the iteration itself. In fact, the number of samples per epoch explicitly depends on $\gamma$, and the step size decreases as $\alpha_n=1/(1+(1-\gamma)n)$.
>
> Regarding the reviewer's specific question, in the average reward setting, the Bellman operator is generally only nonexpansive. This makes it difficult to apply a similar argument due to the crucial role of the discount factor in previous approaches. Nevertheless, we employ a related idea by carefully designing (random) controlled batches to reduce variance. Unfortunately, our approach appears to be constrained to using the square of the span seminorm of $d^k$, as any alternative choice would affect the overall sample complexity.
>
> That being said, it is worth mentioning that in the discounted case, our SAVID+ method can achieve a sample  complexity of $O(SA(1-\gamma)^{-4}\epsilon^{-2})$ for computing an $\epsilon$-optimal policy (see Theorem 4.3). It remains an open question whether a restarting strategy could be devised in this setting to remove the final $1/(1-\gamma)$ factor.
>
> 2. On the connection of our method to Q-learning
>
> The well-known (synchronous version of) Q-learning algorithm takes on a different meaning in the average reward case. In this setting, the goal is to compute bias vectors and (bias) Q-factors to derive optimal policies. The closest analogue to Q-learning in this context is the RVI-Q-learning algorithm [1]. The technical challenge here is that, since the optimal value g* is unknown, an auxiliary function is introduced as an online estimator. However, this approach has a drawback: the resulting operator, while related to the Bellman function, may lose its nonexpansiveness. Nevertheless, it is possible to analyze the RVI-Q procedure using a related iteration, which also turns out to be a stochastic version of the KM iteration (see, for instance, [5] for a nonasymptotic analysis in the unichain case),
> where the set of fixed points is unbounded.
>
> We can now answer the question, which, as mentioned, is central to our motivation. From a purely fixed-point perspective, RVI-Q-learning and similar algorithms can be seen as instances of stochastic KM iterations. It is known that in the absence of noise Halpern iteration is faster with a $1/n$ convergence rate for the fixed-point residual (see references in our paper for further details) whereas KM achieves at best a rate of $1/\sqrt{n}$.
>
> In conclusion, the Reviewer's intuition is fully correct: our goal was precisely to "improve Q-learning" by leveraging the fact that the Bellman error controls the induced policy value (see Proposition 2.1). To reach near-optimal complexity, this idea from fixed-point theory is combined with recursive sampling techniques that have been developed for computing almost-optimal policies
> in average reward MDPs.

---

> > ### Comment · Reviewer_hCuT · 2025-04-02
> >
> > I would like to thank the authors, and I am happy to keep my score.

---

### Official Review · Reviewer_BWFD · 2025-03-14

**Overall Recommendation:** 4

**Summary:**

This paper introduces a novel value iteration algorithm that achieves near-optimal sample complexity in the setting of *weakly communicating* MDP. A weakly communicating MDP is an MDP whose state space is comprised by a set of states that are accessible from one another and an additional set of transient states. The authors consider average reward and discounted cumulative reward value functions.

The algorithm proposed is a value iteration scheme with anchoring. Effectively, it can be seen as a form of Halpern's iteration. The sample complexity is $\tilde{O}(|S||A|\|h^*\|^2/\epsilon^2)$.

The propose 2 algorithms for the average reward case and 2 for the cumulative discounted rewards. In both cases they start with a base algorithm that is incorporating Halpern's iteration on top of value iteration. Yet, as that algorithm relies on knowledge of the Q values from their optimal values, they propose a scheme that uses a doubling trick to be able to call the primary algorithm and while each time getting a better estimate of the optimal Q value.

**Claims And Evidence:**

The claims made are supported by rigorous mathematical proofs.

**Essential References Not Discussed:**

I do not think they have omitted some essential reference.

**Experimental Designs Or Analyses:**

No experiments. The analyses rely on mathematical arguments

**Methods And Evaluation Criteria:**

The paper is of theoretical focus and there are no experiments.

**Other Comments Or Suggestions:**

None

**Other Strengths And Weaknesses:**

Strengths:
* the authors manage to get an almost optimal sample complexity
* the sample complexity is independent of the mixing time of the MDP
* the algorithmic solution requires minor modification over prior method

Weakness:
* no experimental evaluation

**Questions For Authors:**

* What stands in the way of achieving linear dependence on $\| h_* \|_{sp}$?
* Do you think that Halpen iteration could improve with q-learning as well, achieving similar results?

**Relation To Broader Scientific Literature:**

The authors achieve to design an almost optimal algorithm for MDPs that does not depend on the mixing time and the state spain is weakly communicating. It is a very natural RL problem and the algorithm they propose combines value iteration with Halpern's iteration.

**Theoretical Claims:**

I checked the correctness of Proposition 2.1, Proposition 3.1, and Theorem 3.2.

---

> ### Author Rebuttal · Authors · 2025-04-01
>
> We thank the reviewer for the positive comments.
>
> 1. What stands in the way of achieving linear dependence on $\\|h^*\\|_{sp}$?
>
> This is a very interesting and relevant question which we tried to address without success so far. From a technical viewpoint, the quadratic dependence of our sample complexity derives from the batch size $m_k$ in SAVIA which involves the quadratic term $\\|d^k\\|^2$. In order to reduce the sample complexity and reach the theoretical lower bound with linear dependence on $\\|h^*\\|_{sp}$, we considered introducing an additional variance reduction to SAVIA+. A possible approach is to implement a restart similar to the one used in [16] for discounted MDPs, which removed a factor $1/(1-\gamma)$ in the complexity by a carefully designed rerun of the main algorithm. Unfortunately we have not succeeded to adapt this technique to the average reward and at this point it is unclear to us whether this is possible or not. Exploring this is indeed an interesting future direction.
>
> 2. Do you think that Halpern iteration could improve with q-learning?
>
> We are not sure how to interpret this question. It is known that synchronous RVI Q-learning can be viewed as a stochastic version of the Krasnoselskii-Mann (KM) iteration [5, 16]. A discussion on this is also included in the rebuttal to Reviewer hCuT. In the absence of noise, Halpern is known to be optimal achieving a $1/n$ convergence rate [14], whereas KM generally attains at best a $1/\sqrt{n}$ rate [6]. Interestingly, [12] demonstrated that these non-asymptotic rates for KM and Halpern also hold for average MDPs in the tabular setting. In the stochastic framework, KM has already some type of built-in variance reduction so one might conjecture that in combination with Halpern could attain a smaller complexity. Unfortunately, our attempts in this direction did not succeed and produced worse complexity compared to the pure Halpern with recursive sampling presented here.
>
> References
>
> [1] Abounadi J., Bertsekas D., Borkar V.S. (2002), Stochastic approximation for nonexpansive maps: application to Q-learning algorithms, SIAM Journal on Control and Optimization 41(1):1-22.
>
> [2] Azar M.G., Munos R.,  Kappen H.J. (2013), Minimax PAC bounds on the sample complexity of reinforcement learning with a generative model. Machine Learning, 91(3): 325-349.
>
> [3] Bertsekas D.P. (2012), Dynamic Programming and Optimal Control, volume II. Athena Scientific, 4th edition.
>
> [4]  Bravo M., Contreras J.P. (2024), Stochastic Halpern iteration in normed spaces and applications to reinforcement learning. arXiv:2403.12338.
>
> [5] Bravo M., Cominetti R., (2024) Stochastic fixed-point iterations for nonexpansive maps: Convergence and error bounds, SIAM Journal on Control and Optimization, 69:191-219.
>
> [6] Contreras J.P., Cominetti R., (2022), Optimal error bounds for nonexpansive fixed-point iterations in normed spaces, Mathematical Programming, 199(1-2):343-374.
>
> [7]  Ganesh S., Mondal W.U., Aggarwal V. (2024), Order-Optimal Global Convergence for Average Reward Reinforcement Learning via Actor-Critic Approach, arXiv:2407.18878v2.
>
> [8] Jin Y., Gummadi R., Zhou Z., Blanchet J. (2024a) Feasible Q-learning for average reward reinforcement learning. International Conference on Artificial Intelligence and Statistics.
>
> [9] Jin Y., Sidford A. (2020), Efficiently solving MDPs with stochastic mirror descent. International Conference on Machine Learning.
>
> [10] Lieder F. (2021), On the convergence rate of the Halpern-iteration. Optimization Letters, 15(2):405-418.
>
> [11] Lee J., Ryu E. (2023), Accelerating value iteration with anchoring. Neural Information Processing Systems.
>
> [12] Lee J., Ryu E. (2025), Optimal non-asymptotic rates of value iteration for average-reward MDPs. International Conference on Learning Representations.
>
> [13] Puterman M.L. (2014) Markov Decision Processes: Discrete Stochastic Dynamic Programming. John Wiley and Sons, 2nd edition.
>
> [14] Sabach S., Shtern S. (2017), A first order method for solving convex bileveloptimization problems. SIAM Journal on Optimization, 27(2):640-660.
>
> [15] Tuynman A., Degenne R., Kaufmann E. (2024), Finding good policies in average-reward Markov decision processes without prior knowledge. Neural Information Processing Systems.
>
> [16] Wainwright M.J. (2019). Variance-reduced-learning is minimax optimal. arXiv:1906.04697
>
> [17] Wainwright M.J. (2019). Stochastic approximation with cone-contractive operators: Sharp $\ell_\infty$ bounds for Q-learning. arXiv:1905.06265
>
> [18] Wang S., Blanchet J., Glynn P. (2024), Optimal sample complexity for average reward Markov decision processes. International Conference on Learning Representations.
>
> [19] Zurek M., Chen Y. (2024), Span-based optimal sample complexity for weakly communicating and general average MDPs. Neural Information Processing Systems.
>
> [20] Zurek M., Chen Y. (2025), The Plug-in Approach for Average-Reward and Discounted MDPs: Optimal Sample Complexity Analysis. arXiv:2410.07616.

---

### Official Review · Reviewer_jzY5 · 2025-03-14

**Overall Recommendation:** 4

**Summary:**

This paper studies the sample complexity of weakly communicating average-reward MDPs assuming access to a generative model. The authors focus on developing model-free algorithms by using a stochastic version of Halpern iteration. They show that this approach achieves a sample complexity bound in terms of the span of a bias vector solving the Bellman optimality equations, and unlike some prior work, their algorithm does not require any prior knowledge of this complexity parameter. They achieve this by using a doubling trick combined with a stopping condition. They also adapt their algorithm to discounted MDPs.

## update after rebuttal

I think the authors for their response. I'll maintain my score.

**Claims And Evidence:**

The main claims of this paper are all of a theoretical nature and supported by proofs.

**Essential References Not Discussed:**

Discussion on Jin et al. 2024b and https://arxiv.org/abs/2410.07616v1 should be added. See "Relation to Broader Scientific Literature" above.

here are several missing references all related to termination conditions. Propositions 2.1 and 4.1 are very standard and well-known (e.g. I think they appear in the Bertsekas Dynamic Programming and Optimal Control books). A stopping rule guaranteeing near-optimality is provided in Tuynmann et al. (2024) (the authors cite this paper but do not mention this stopping rule, which seems at least closely related to the one used in this paper, and so discussion of their relationship seems needed).

**Experimental Designs Or Analyses:**

N/A

**Methods And Evaluation Criteria:**

The methods, namely stochastic Halpern iteration, make much sense for this problem. The evaluation criterion of sample complexity to learn near-optimal policies is standard for this well-studied problem. Measuring the sample complexity required to find a point with small Bellman error in the discounted setting (Theorem 4.2) is new. While this is an interesting finding which I do think is worthy of inclusion in the paper, it is unclear to me what use this has (beyond the fact that low Bellman error implies a bound on the suboptimality, which is probably what we actually care about).

Related to this task of finding a point with small Bellman error, regarding the comments made about prior work on approximately line 426, first column, it does not follow that these works require the stated complexity $SA/((1-\gamma)^3 \varepsilon^2)$ to obtain an $\varepsilon$ Bellman residual error, it just means that their sample complexity for doing so is upper-bounded by this amount. In fact, several prior works achieve a sampling complexity of $SA/((1-\gamma)^2 \varepsilon^2)$ [Wang et al 2023, Zurek & Chen 2024] for computing an $\epsilon$-optimal policy. In light of this and my comment above on Bellman error vs. suboptimality, I have reservation with the claim on line 429 on Theorem 4.2 being the best known complexity.

**Other Comments Or Suggestions:**

N/A

**Other Strengths And Weaknesses:**

The paper is generally written clearly.

Since the results for discounted MDPs are basically half of the paper, the authors should discuss more related work for solving discounted MDPs (such works are only given a few lines in the introduction).

The authors should also discuss more related work on model-based methods and include the comparison in Table 1. Particularly in the tabular setting, the distinction between model-based and model-free methods are blurry and not particularly useful.

In the introduction (around line 49 column 2) the authors seem to claim that their main contribution is that they are the first to apply a Halpern technique to average-reward MDPs in the generative model setting. However in line 207 column 1 the authors cite the earlier work Bravo and Cominetti (2024) and explicitly mention it was used in this same problem setting. Therefore it seems like it would be beneficial for the authors to clarify their main contributions.

**Questions For Authors:**

What is the use of a Bellman error bound (Theorem 4.2) beyond the fact that low Bellman error implies a bound on the suboptimality?

**Relation To Broader Scientific Literature:**

The achieved sample complexity is inferior to the best prior model-based approaches. The achieved sample complexity is slightly better than that of the best prior model-free approach of Zhang and Xie (2023), removing a lower order ($O(1/\varepsilon)$) term present in their bound. There also exists an earlier model-based approach https://arxiv.org/abs/2410.07616v1, which is not cited in this paper, which achieves a similar sample complexity bound also without prior knowledge of the span parameter. Therefore the main contribution of this paper is that it is simultaneously model-free, prior-knowledge-free, and obtains a sample complexity bound which is suboptimal by only one factor of the span parameter. This is still a nice contribution, in particular because the problem of removing the need for prior knowledge is important and challenging.

The results for discounted MDPs do not seem to offer significant improvements upon prior work so I am uncertain about their value. (For instance, the dependence on $Q^\star$, obtained in Theorem 4.3, is also obtained in Jin et al. 2024b, which is not mentioned. I think more discussion about the relationship of the discounted MDP results to the literature would improve the paper.

**Theoretical Claims:**

I looked over most results for the average-reward setting, which seem correct.

---

> ### Author Rebuttal · Authors · 2025-04-01
>
> We thank the reviewer's constructive feedback. For the list of references please refer to the reply to review BWFD.
>
> 1. Advantage of SAVIA+ compared to [8] and [20]
>
> The papers [8] and [20] propose model-free and model-based methods under the mixing time and weakly communicating assumptions, respectively. Although they can run without prior-knowledge of mixing times and bias span, they do not allow to determine the number of iterations required to achieve an $\epsilon$-optimal policy, unless one has a priori bounds on $t_{mix}$ or $h^*$. While our SAVIA algorithm has the same drawback (see Section 3.2), by using a doubling trick and the empirical Bellman error we get SAVIA+ which finds an $\epsilon$-optimal policy with high probability. This is a key advantage over previous methods, and we believe it is the first method that is "fully" prior-knowledge-free. We will clarify this in the revised version.
>
> 2. Novelty of results for discounted MDPs (DMDPs)
>
> Our main contribution is clearly the sample complexity for average MDPs. However, given the novelty of anchoring and recursive sampling in RL, exploring their use for  DMDPs is natural and instructive. Let us discuss the relevance of our Bellman error by comparing to [18, 19]. For ergodic DMDPs with $t_{mix} \le 1/(1-\gamma)$, [18] obtains sample complexity $O(SA t_{mix}(1-\gamma)^{-2} \epsilon^{-2})$. For weakly communicating and general MDPs, if $H ,B+H \le 1/(1-\gamma)$, [19] obtains $O(SA H(1-\gamma)^{-2} \epsilon^{-2})$ and
> $O(SA (B+H)(1-\gamma)^{-2} \epsilon^{-2})$, respectively. These bounds include the factors $t_{mix}, H, B$ and hold only for $\gamma$ large enough. In contrast our Theorem 4.2 provides a sample complexity $O(SA (1-\gamma)^{-2} \epsilon^{-2})$ to achieve $\epsilon$-Bellman error, with no additional parameters and assumptions. We believe this is still a meaningful contribution.
>
> The fixed-point residual, corresponding to the Bellman error in MDPs, has been widely used as a performance measure [10, 14] and, unlike the distance to the Q*, it is computable. This also holds in the generative setting where one can compute the empirical Bellman error, whereas estimating the policy error is more challenging as it involves the g*. Moreover, the information-theoretic lower bound of the Bellman error has been recently studied in the tabular setup [11, 12]. In the generative setting the lower bound for $\epsilon$-optimal policy is $O(SA (1-\gamma)^{-3} \epsilon^{-2})$ [2]. Our Theorem 4.2 shows that the Bellman error is at most $O(SA(1-\gamma)^{-2}\epsilon^{-2})$, which we believe is the best upper bound currently available and does not follow from previous results. We will clarify this in the revision.
>
> Regarding Theorem 4.3, as noted by the reviewer, [20] presents a model-based approach with complexity depending on V*.  Although [20, Theorems 9 and 10] guarantee the convergence, they require knowledge of V* to determine the number of iterations to obtain an $\epsilon$-optimal policy. In contrast, SAVID+ does not require knowledge of Q* to get $\epsilon$ optimality. In this sense, our Theorem 4.3 improves over prior results. As suggested, we will expand our review of prior works for DMDPs, with a more detailed discussion on model-based and model-free methods.
>
> 3. Propositions 2.1 and 4.1
>
> For completeness and readability we think it is useful to include Propositions 2.1 and 4.1 which connect the Bellman error to the policy error, justifying our focus on the former. Proposition 2.1 is a Q-factor variant of a result for bias vectors in [13], whereas Proposition 4.1 is a simple consequence of estimates for contractions. However, we could not find these statements explicitly in [3] nor other prior works. We will clarify this in the revision and would be happy to include a reference for these facts.
>
> 4. Stopping rule
>
> Thanks for mentioning the connection with [15]. We note however that [15] considers a model-based algorithm with a stopping rule based on an estimate of the diameter of the MDP, restricting the framework to communicating MDPs. In the revision we will discuss this connection.
>
> 5. Prior work on stochastic Halpern in MDPs
>
> We are sorry for the misunderstanding (line 49 col. 2). We intended to stress that our work is the first to combine recursive sampling with Halpern iteration. As the reviewer noted, we acknowledged that [4] used Halpern iteration in this context (line 207). Specifically, [4] considers stochastic Halpern iteration for nonexpansive maps using minibatches, establishing error bounds in expectation. For average MDPs this yields a residual of at most $\epsilon$ in expectation with a larger sample complexity $O(SA\epsilon^{-7})$. We suspect it should be possible to get $\epsilon$-optimality in high probability with that rate, but the bound is not explicit in its dependence on the $H$. So it is unclear whether it can be implemented without prior knowledge (as the case of [8, 20]). We will clarify this in the revised version.

---

### Official Review · Reviewer_mUiL · 2025-03-14

**Overall Recommendation:** 3

**Summary:**

The established sample complexity of $O(\frac{1}{\epsilon^2})$ for average reward MDPs using a interesting approach of Halpern's iteration. The result is independent of mixing time which is very important.

**Claims And Evidence:**

Looks good.

**Essential References Not Discussed:**

> The paper [1], also has similar results $O(\frac{1}{\epsilon^2})$ convergence using span norm. Requesting authors to compare their work with this.


> [2] also proves sample complexity of $O(\frac{1}{\epsilon^2})$, without knowledge of mixing time.




[1] @misc{zurek2024spanbasedoptimalsamplecomplexity,
      title={Span-Based Optimal Sample Complexity for Average Reward MDPs},
      author={Matthew Zurek and Yudong Chen},
      year={2024},
      eprint={2311.13469},
      archivePrefix={arXiv},
      primaryClass={cs.LG},
      url={https://arxiv.org/abs/2311.13469},
}

[2] @misc{ganesh2024orderoptimalglobalconvergenceaverage,
      title={Order-Optimal Global Convergence for Average Reward Reinforcement Learning via Actor-Critic Approach},
      author={Swetha Ganesh and Washim Uddin Mondal and Vaneet Aggarwal},
      year={2024},
      eprint={2407.18878},
      archivePrefix={arXiv},
      primaryClass={cs.LG},
      url={https://arxiv.org/abs/2407.18878},
}

**Experimental Designs Or Analyses:**

No

**Methods And Evaluation Criteria:**

Seems so.

**Other Comments Or Suggestions:**

Suggestions:  Better literature surevey would better position the paper and its contribution.

**Other Strengths And Weaknesses:**

Strengths: Paper is well written as easy to read.   The theoretical results are convincing. The approach (Halpern's iteration) used is very interesting.

Weakness: It is not sure, how the result improves the SOTA, as the $O(\frac{1}{\epsilon^2})$ sample complexity without knowledge of mixing time already exits in [1].



 [1] @misc{ganesh2024orderoptimalglobalconvergenceaverage,
      title={Order-Optimal Global Convergence for Average Reward Reinforcement Learning via Actor-Critic Approach},
      author={Swetha Ganesh and Washim Uddin Mondal and Vaneet Aggarwal},
      year={2024},
      eprint={2407.18878},
      archivePrefix={arXiv},
      primaryClass={cs.LG},
      url={https://arxiv.org/abs/2407.18878},
}

**Questions For Authors:**

Q1: The paper states the average reward MDP has no contraction mapping, hence uses Halpern iteration (anchored VI) that has convergece rate of $O(\frac{1}{k})$. However, as shown in section 6.6 of [1], it proves  average reward MDP Bellman operator is a $\gamma$-contraction where $\gamma$ is a function of mixing time of the MDP. It would be appreciated, if the authors could clarify these two.






[1] [Wiley Series in Probability and Statistics] Martin L. Puterman - Markov decision processes_ discrete stochastic dynamic programming (1994, Wiley-Interscience)

**Relation To Broader Scientific Literature:**

Average reward MDPs are crucial field of studies, with many real world setting, hence the sample complexity for the same, is of great importance.

**Theoretical Claims:**

I didn't verify the proofs.

---

> ### Author Rebuttal · Authors · 2025-04-01
>
> We thank the reviewer for the opportunity to clarify some points that were not transparent
> in our manuscript, and to better put in perspective our contribution with respect to
> previous work. For the list of references please refer to the reply to Reviewer BWFD.
>
> 1. Comparison with [19]
>
> There are two major differences between our results and those in [19, Zurek & Chen].
> First, in contrast with SAVIA+ which is model-free, the algorithm proposed in [19] is
> model-based with its higher memory requirements for storing the empirical transition kernel.
> Second, and more importantly, Zurek & Chen's method requires prior knowledge of an upper
> bound for the span seminorm of the bias vector $H\geq \\|h^*\\|_{sp}$, whereas SAVIA+ does not
> require any prior-knowledge.
>
> 2. Comparison with [7]
>
> Ganesh et al. [7] study a model-free algorithm for average reward MDPs, although in a  different context of Markovian sampling and function approximation which allows to deal with large state spaces by considering a parameterized family of policies. The proposed actor-critic algorithm is designed to optimize  the parameters of the policy. The results are restricted to ergodic MDPs, although their method does not require prior-knowledge of the mixing time. The main result, Theorem 1, establishes an optimal asymptotic convergence rate in expectation, although no finite error bounds in high probability are provided. As a consequence the model and its underlying assumptions, as well as the algorithm and the complexity results, are of a different nature compared to our paper and neither one implies the other.
>
>
>
> 3. Strict contraction vs nonexpansivity of Bellman's operator
>
> For average reward MDPs with finite mixing times  Bellman's operator is indeed a strict contraction in span seminorm. However, this property fails in the weakly communicating case where $t_{mix}$ can be infinite, and Bellman's operator may be just nonexpansive with multiple fixed points. Consider a simple example with two states $S=\\{s_1,s_2\\}$ and two actions $A=\\{a_1,a_2\\}$ with deterministic transitions: action $a_1$ keeps the process at the current state, whereas $a_2$ moves to the other state; all rewards are $1$ except when moving from $s_2$ to $s_1$ under action $a_2$ whose reward is $0$. The optimal reward is $g^*=1$ and Bellman's operator for the bias $h=(h_1,h_2)$ is (see line 93 second column of our paper): $T(h_1,h_2)=(\max\\{h_1,h_2\\},\max\\{h_1-1,h_2\\})$ which is nonexpansive in span seminorm but not a contraction. In fact $T$ has a continuum of fixed points even up to identification modulo constants, namely, all vectors $h=(h_1,h_2)\in\mathbb{R}^2$ such that $h_1-h_2 \in[0,1]$.

---

### Decision · Program_Chairs · 2025-05-01

**Decision:**

Accept (poster)

**Comment:**

The paper introduces a model-free algorithm to identify, in the generative model setting, $\epsilon$ optimal policies in weakly communicating MDPs. The algorithm is a value iteration procedure with anchoring that can be seen as a form of Halpern’s iteration. The authors establish an upper bound on its sample complexity, scaling as $SA\| h^\star\|^2/\epsilon^2$, where $\| h^\star\|$ is the span semi-norm of the bias vector (in the average reward scenario). The algorithm does not require any a-priori knowledge. In summary, the main contribution of this paper is to devise an algorithm simultaneously model-free, prior-knowledge-free, and to obtain a sample complexity bound which is suboptimal by only one factor of the span parameter.

All reviewers acknowledge the quality of the analysis. The only concerns pertain to the novelty of the approach and the comparison with existing work. Halpern’s iteration has been used previously for learning in average reward MDPs. There are also related work presenting algorithms with similar sample complexity but with the knowlegde of an upper bound on the span. The authors here remove this knowledge (by using a doubling trick).
No experiments are presented and the results are purely theoretical.